# Intrinsic monitoring of learning success facilitates memory encoding via the activation of the SN/VTA-Hippocampal loop

Pablo Ripollés[1,2], Josep Marco-Pallarés[1,2], Helena Alicart[1], Claus Tempelmann[3], Antoni Rodríguez-Fornells[1,4*†], Toemme Noesselt[5,6*†]

[1]Cognition and Brain Plasticity Group, Bellvitge Biomedical Research Institute - IDIBELL, Barcelona, Spain; [2]Department of Basic Psychology, University of Barcelona, Barcelona, Spain; [3]Department of Neurology, Otto-von-Guericke University, Magdeburg, Germany; [4]Catalan Institution for Research and Advanced Studies, ICREA, Barcelona, Spain; [5]Department of Biological Psychology, Otto-von-Guericke-University Magdeburg, Magdeburg, Germany; [6]Center for Behavioral Brain Sciences, Magdeburg, Germany

*For correspondence: antoni.rodriguez@icrea.cat (AR-F); toemme@med.ovgu.de (TN)

†These authors contributed equally to this work

Competing interests: The authors declare that no competing interests exist.

**Abstract** Humans constantly learn in the absence of explicit rewards. However, the neurobiological mechanisms supporting this type of internally-guided learning (without explicit feedback) are still unclear. Here, participants who completed a task in which no external reward/feedback was provided, exhibited enhanced fMRI-signals within the dopaminergic midbrain, hippocampus, and ventral striatum (the SN/VTA-Hippocampal loop) when successfully grasping the meaning of new-words. Importantly, new-words that were better remembered showed increased activation and enhanced functional connectivity between the midbrain, hippocampus, and ventral striatum. Moreover, enhanced emotion-related physiological measures and subjective pleasantness ratings during encoding were associated with remembered new-words after 24 hr. Furthermore, increased subjective pleasantness ratings were also related to new-words remembered after seven days. These results suggest that intrinsic—potentially reward-related—signals, triggered by self-monitoring of correct performance, can promote the storage of new information into long-term memory through the activation of the SN/VTA-Hippocampal loop, possibly via dopaminergic modulation of the midbrain.

## Introduction

In the last decade, multiple studies have shown that external rewards can enhance human learning and memory through Long Term Potentiation (LTP) processes triggered by the co-activation of a dopamine-dependent loop formed by the ventral striatum (VS), the substantia nigra/ventral tegmental area complex (SN/VTA), and the hippocampus (HP; hereafter referred to as the SN/VTA-HP loop; *Lisman and Grace, 2005*; *Goto and Grace, 2005*; *Lisman et al., 2011*; *Shohamy and Adcock, 2010*). According to this model, the activation of this reward-memory loop starts when new information that needs to be stored in memory 'arrives' at the HP. A signal is then sent to the SN/VTA through the VS, which is thought to integrate affective, motivational, and goal-directed information into the loop (*Lisman and Grace, 2005*; *Goto and Grace, 2005*). SN/VTA neurons are then disinhibited, facilitating dopaminergic signalling into the HP (*Lodge and Grace, 2006*; *Shohamy and Adcock, 2010*), which in turn enhances memory formation. In accord with this model, recent

**eLife digest** Research shows that a reward such as money, or even simply the promise of such a reward, can boost the formation of long-term memories. However, in our everyday lives, we continually gain new knowledge and make new memories in the absence of any obvious immediate reward.

Rewards activate a network of brain regions that includes the hippocampus, which has a key role in memory, plus several areas that release the chemical messenger dopamine, which boosts memory formation. However, it was not clear whether this network of brain regions also supports learning that is driven internally rather than by external rewards or incentives.

Ripollés et al. have now tested this idea by asking thirty-six volunteers to try and learn the meaning of new words by reading pairs of sentences, all while lying down inside a brain scanner. Half of the paired sentences provided a clear and obvious meaning for the new word. As such, the volunteers were reasonably aware when they'd learned the meaning of a new word without any external feedback. This approach confirmed that the activity of the brain's reward-memory loop did indeed increase whenever a volunteer learned a new word.

Next, outside the brain scanner, the volunteers performed the same task but this time they had to rate how engaging and enjoyable they found it after each trial. Emotional responses such as enjoyment trigger sweating, which alters the electrical activity of the skin. Ripollés et al. observed greater changes in this "electrodermal" activity when the volunteers learned words that they would go on to remember one day later, than when they learned words that they would quickly forget. The volunteers also reported greater enjoyment when learning the words that they would subsequently remember better, even after seven days.

Overall, these findings suggest that internally driven learning is in itself rewarding, and that under certain circumstances at least it can activate the brain's reward-memory circuit. A key question for the future is whether tapping into intrinsically rewarding forms of learning might be a more effective educational strategy than relying on external feedback and incentives. This could be crucial to improving the design of educational programs – for example, in teaching literacy and foreign languages – and for improving the recovery of verbal skills lost after stroke.

evidence suggests that both the anticipation of an explicit reward (e.g., the possibility to win money; *Adcock et al., 2006*; *Wittmann et al., 2005*; *Wolosin et al., 2012*; *Callan et al., 2008*) and also intrinsic motivational states (e.g., curiosity about future events; *Gruber et al., 2014*) can modulate memory formation through the activation of the SN/VTA-HP loop.

Whereas these previous studies manipulated participants' expectations (i.e., money/curiosity) in order to enhance memory, in everyday life we constantly engage in learning processes in which the possibility to obtain an external reward is mostly absent. If neither extrinsic feedback regarding the learning success nor external reward is given, some internal system must still decide whether the output of the learning experience (i.e., new information) is valid and should thus be stored into long-term memory. One could predict that successful internally-guided learning (i.e., learning without explicit feedback) might trigger intrinsic reward-related processes after positive monitoring (related to a feeling of *efficacy*; *White, 1959*), favouring the encoding of the learning episodes into the memory system. Concordantly, psychological theories on self-regulated learning have proposed that internal monitoring and trial-by-trial evaluation of learning success may be crucial determinants of learning in complex environments (*Bjork et al., 2013*). Considering the importance of the SN/VTA-HP loop in the regulation of extrinsically motivated learning (*Adcock et al., 2006*; *Wittmann et al., 2005*; *Callan et al., 2008*; *Wolosin et al., 2012*; *Gruber et al., 2014*), we hypothesized that the recruitment of this reward-memory loop should be instrumental in storing successful learning experiences also in the absence of explicit feedback, especially after positive intrinsic evaluation of the learning outcome.

Related to this, we recently developed a learning task (*Ripollés et al., 2014*; see also *Mestres-Missé et al., 2007*) that mimicked our capacity to learn the meaning of new-words presented in verbal contexts, a process that usually occurs without external guidance and that is considered one of

the most important sources of vocabulary learning during childhood years (*Nagy et al., 1985*). Indeed, there is an extensive body of evidence showing that children (as young as eight years-old) can learn new words from written contexts (e.g., by reading) on their own, not only when explicitly instructed to do so, but also during incidental learning conditions (*Jenkins et al., 1984*; *Konopak et al., 1987*; *Kuhn and Stahl, 1998*; *Nagy et al., 1985*, *1987*; *Werner and Kaplan, 1950*). Moreover, as adult learners, we also constantly face the problem of learning the meaning of new-words in our own native or non-native languages, and in most cases, the repeated presentation of a new-word in different verbal contexts can allow to discover the meaning of the new-word (*Nation, 2001*). Therefore, our word-learning task is ideally suited to test internally-guided learning as: (i) in our task participants are able to learn the meaning of artificially created new-words by using contextual information, without the need for explicit feedback or reward; and (ii) our paradigm mimics a learning process that occurs in real-world environments. Importantly, we recently showed that, in our paradigm, successful meaning extraction enhanced fMRI-signals within the VS. Moreover, this activity was not related to novelty, attention or exertion of effort (*Ripollés et al., 2014*). While our previous results suggested that learning the meaning of a new word triggered reward-related signals within the VS, we never assessed the effect that these internally elicited signals had on the formation of longer-lasting memory traces. Thus, an important question arises: can positive monitoring of learning success—in the absence of explicit feedback or external reward—mediate the entrance of new information into long-term memory via the modulation of the SN/VTA-HP loop? In order to address this question, we first re-analyzed the functional magnetic resonance imaging (fMRI) data from our previous work (Exp. 1) using a region-of-interest (ROI) analysis that focused on all the areas of the SN/VTA-HP loop (constrained to be reward-related by means of a meta-analysis on reward; NeuroSynth, *Yarkoni et al., 2011*). In addition, we used a post-scan retrieval test to assess the possible memory benefits induced by the activation of the SN/VTA-HP loop during successful encoding. We hypothesized that increased brain activity and functional connectivity within the areas of the loop, in the absence of any external reward, should be associated with enhanced memory formation (i.e., greater activity and connectivity during encoding for later remembered vs. later forgotten items).

Furthermore, considering that the effects of dopamine are stronger during the late stage of LTP (*Murayama et al., 2014*; *Adcock et al., 2006*; *Gruber et al., 2014*), we carried out a second experiment using a modified version of our word-learning task (**Exp. 2**) in which an additional surprise recognition test was carried out 24 hr after encoding. Based on our previous findings suggesting a strong relation between language learning and reward processing (*Ripollés et al., 2014*) and also on studies showing that uncovering the solution to a problem is closely tied to an increase in subjective pleasantness (*Kizilirmak et al., 2015*; *Bechara and Damasio, 2005*), we recorded subjective self-reported ratings of arousal and pleasantness during the encoding phase of our task, along with objective physiological measures (electrodermal activity). Indeed, evidence suggests that emotion-related signals—assisting cognitive processes such as learning and decision making—can be captured by electrodermal activity (EDA; *Bechara and Damasio, 2005*). In particular, skin conductance responses (SCRs) have been linked to enhanced memory formation, motivational behavior (*Cahill et al., 1998*), explicit reward-related processes (*Lole et al., 2014*; *Mas-Herrero et al., 2014*), and also to a modulation of the orbitofrontal cortex, the amygdala, and the striatum (*Critchley et al., 2000*). Finally, a third experiment (**Exp. 3**) was carried out to replicate the behavioural effects of Exp. 2 over longer retention intervals (one week). We hypothesized that if internal monitoring of learning success would indeed trigger reward-related processes that ultimately enhance memory formation, new-words remembered after longer retention periods should be associated with increased pleasantness ratings and enhanced SCRs during the initial encoding phase.

## Results

### Does successful internally-guided learning activate the entire reward-related SN/VTA-Hippocampal loop?

To answer this question, 36 participants completed an fMRI version of our word-learning task (see *Figure 1A*), in which the meaning of a new-word could be learned from the context provided by two sentences built with an increasing degree of contextual constraint (*Mestres-Missé et al., 2010*).

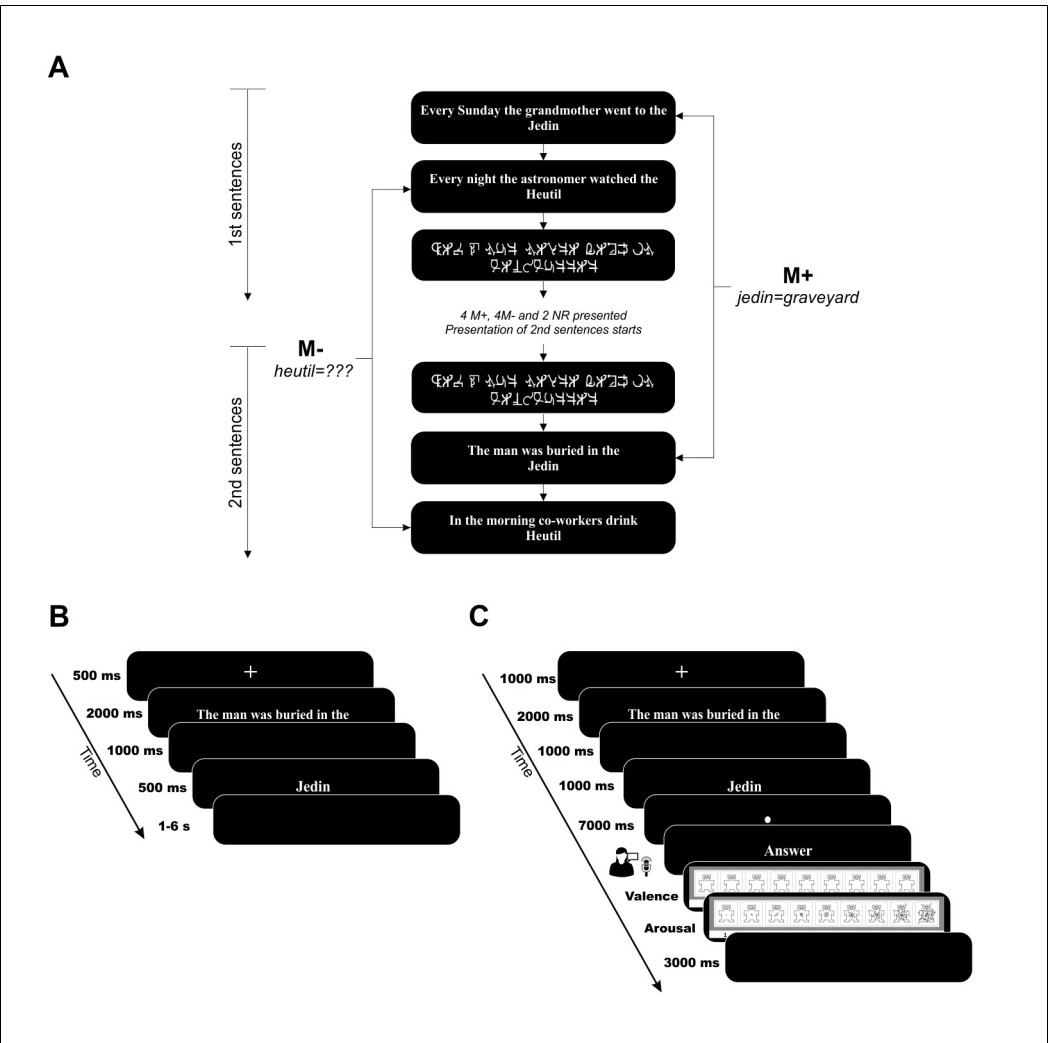

**Figure 1.** Schematic overview of trials and conditions in the word-learning paradigm. (**A**) Participants completed 10 short encoding blocks. Four pairs of sentences of each condition (M+, M-) and two pairs of non-readable sentences (NR, only for the fMRI experiment) were presented per block. Note that first sentences for each condition are always presented prior to and in a different order than second sentences. (**B**) Each trial in the fMRI experiment started with a fixation cross lasting 500 ms followed by 6 German words of the sentence for 2 s and 1 s of dark screen. Finally, the new-word was presented for 500 ms. Between trials, there was a variable inter-trial interval of 1 to 6 s (Poisson distribution, *Hinrichs et al., 2000*). (**C**) Each trial in Exp. 2 started with a fixation cross lasting 1000 ms, continued with the 7 first Spanish words of the sentence presented for 2 s, and was followed by a 1 s duration dark screen. The new-word was presented for 1000 ms. and was followed by 7 s of a small fixation point presented in the middle of the screen. For first sentences, a new trial was presented after 3 s of dark screen. For second sentences, after this period, a screen with the word *Answer* appeared and subjects had 3 s to produce a verbal answer. Then, the SAM scales for pleasantness and arousal were sequentially presented (the experiment did not continue until participants provided a rating). Finally, a new second sentence trial started after 3 s of dark screen. M+ (meaning extraction possible during second presentation); M- (correct meaning extraction not possible during second presentation); NR sentences (non readable).

Only half of the pairs of sentences disambiguated multiple meanings, allowing the encoding of a congruent meaning of the new-word during its second presentation (M+ condition; see Material and methods). For the other pairs, the new-word was not associated with a congruent meaning across the sentences, and could not be learned (M- condition). In addition, non-readable sentences (NR) were presented as a control. During encoding, participants recognized the correct meaning of

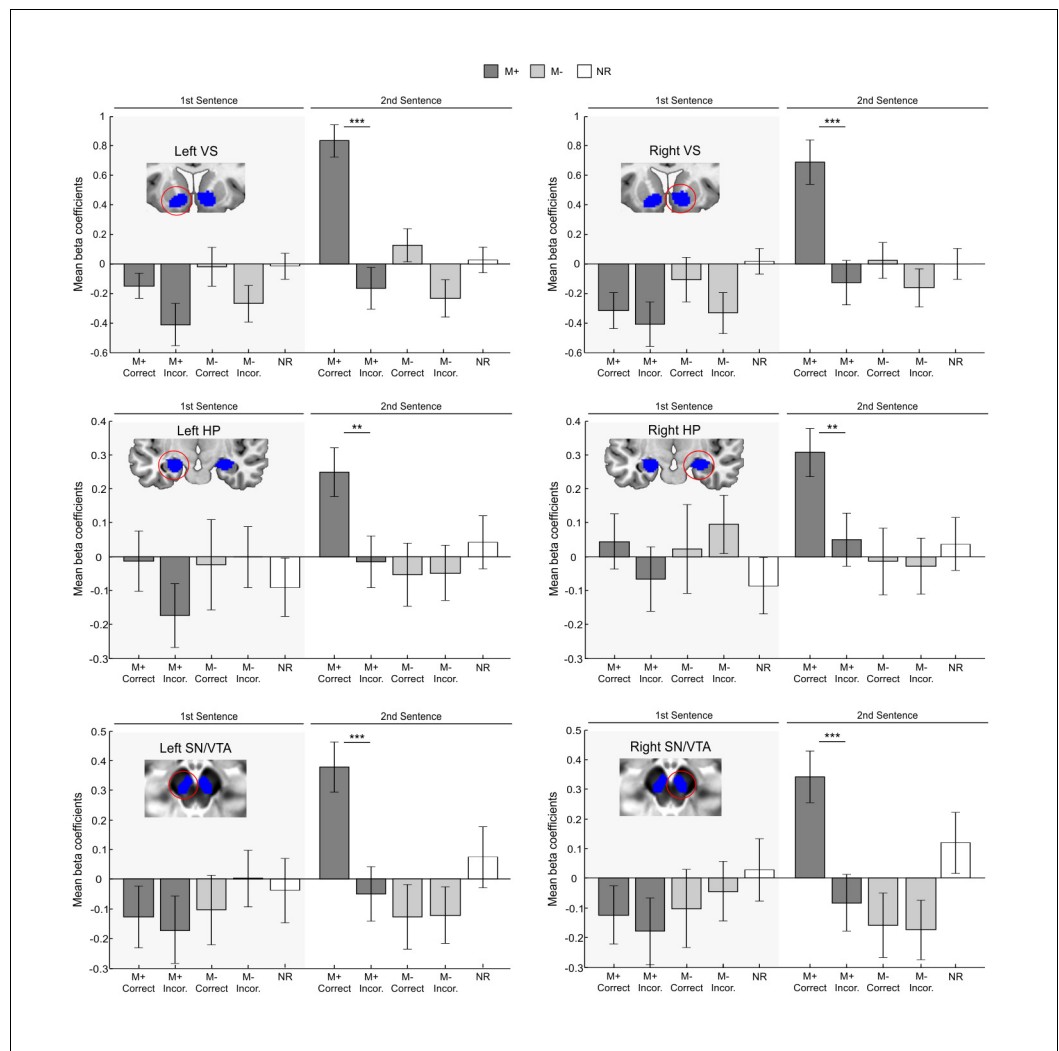

**Figure 2.** ROI analysis controlling for novelty and task difficulty. Blue areas depict independent ROIs used for beta-extraction overlaid on a canonical T1-weighted template (VS, HP) or the mean proton density normalized template from all subjects (SN/VTA). Bar graphs show mean beta coefficients within ROI for each condition of interest (M+ correct first sentence, M+ incorrect first sentence, M- correct first sentence, M- incorrect first sentence, M+ correct second sentence, M+ incorrect second sentence, M- correct second sentence, M- incorrect second sentence; the NR condition is shown as a control) with standard error of the mean (dark grey for M+; light grey for M-; white for NR). Paired t-test comparisons for all correct versus incorrect conditions revealed significant differences in all ROIs only for M+ correct versus incorrect trials during the second sentence presentation, when participants successfully learned the meaning of a new-word. L, Left Hemisphere; M+, congruent meaning extraction possible; M-, congruent meaning extraction impossible; NR, non-readable sentences; VS, Ventral Striatum; HP, Hippocampus; SN/VTA, Substantia Nigra/ Ventral Tegmental Area. ***p<0.001; **p<0.005.

(mean, std) 60 ± 15% of M+ new-words and correctly indicated an absence of coherent meaning in 61 ± 22% of M- cases (see Appendix 1 for more details). Given our explicit a-priori hypothesis regarding the VS, HP, and SN/VTA, an ROI analysis was performed. In order to avoid circularity, the ROIs were created using independent data and were restricted to those regions related to reward-related processes (see Materials and methods).

In addition to reward, several structures, including the VS and the SN/VTA, are also activated by the novelty or salience of the stimuli (*Guitart-Masip et al., 2010*; *Bunzeck and Düzel, 2006*), by attentional processes, by task-difficulty/exertion of effort (*Boehler et al., 2011*), or by the response accuracy itself—regardless of successful meaning extraction (i.e., correct M- trials). The design of our

paradigm allows us to address these alternative explanations by including the incongruent condition (M-, no learning) and the order of presentation (first or second sentence) in our analyses (note that participants were equally prompted to complete the task for both M+ and M- conditions). Regarding the possible effort-related interpretation of the SN/VTA activation (*Boehler et al., 2011*), previous studies using a similar paradigm have shown that incongruent conditions (M-) can be more difficult and effortful to resolve than congruent ones (M+; e.g. *Mestres-Missé et al., 2014*). Finally, the inclusion of the M- condition and the order of presentation also account for possible novelty and accuracy or subjective meaning confounds, as: (i) new-words are equally novel to participants for M+ and M- conditions; (ii) novelty effects should be more apparent during the first than during the second presentation of a particular new-word; (iii) generalized effects of accuracy or subjective meaning attribution should be apparent during correct rejection or false alarms of M- trials.

In order to rule out all aforementioned alternative explanations, a full-factorial ROI analysis was performed (for a similar approach, see e.g., *Adcock et al., 2006*; *Gruber et al., 2014*). Individual beta coefficients for each participant were extracted from the fMRI word-learning task and submitted to a $2 \times 2 \times 3 \times 2 \times 2$ repeated measures analysis of variance (ANOVA) with the factors Hemisphere (Left, Right), Order (first sentence, second sentence), ROI (VS, HP, SN/VTA), Condition (M+, M-), and Response (Correct, Incorrect). We then tested whether the effects revealed during encoding in the HP, VS, and SN/VTA really reflected signal changes due to successful meaning extraction or rather to unspecific effects of accuracy, second word presentation, or experimental condition. Corroborating our hypothesis, we found a significant interaction of Order $\times$ Condition $\times$ Response [$F(1,35) = 4.254$, $p<0.047$, partial $\eta2 = 0.108$]. Moreover, this triple interaction was further affected by region [$F(2,70) = 4.258$, $p<0.018$, partial $\eta2 = 0.108$; the effect is somewhat smaller for the HP ROIs, but still significant; see paired t-tests below] but not by hemisphere. This interaction reflects greater activation for correct versus incorrect trials only for second sentences and for the M+ condition. Importantly, there was no main effect of condition ($p>0.19$), further emphasizing that the reported effects were driven by the M+ correct trials. Concordantly, paired t-test comparisons for all correct versus incorrect conditions showed significant differences in all ROIs only for M+ correct versus incorrect trials (see *Figure 2*) during the second sentence presentation [left VS, $t(35) = 6.29$, $p<0.001$, $d = 1.31$; left HP, $t(35) = 3.59$, $p<0.002$, $d = 0.59$; left SN/VTA, $t(35) = 4.50$, $p<0.001$, $d = 0.79$; right VS, $t(35) = 4.73$, $p<0.001$, $d = 0.89$; right HP, $t(35) = 3.28$, $p<0.003$, $d = 0.56$; right SN/VTA, $t(35) = 4.25$, $p<0.001$, $d = 0.76$; two-tailed, at $p<0.05$ FDR-corrected]. In addition, no significant Order $\times$ Condition $\times$ Response [$F(1,35) = 0.354$, $p>0.55$, partial $\eta2 = 0.010$] interaction was found when using a control ROI based at the primary visual cortex (see Materials and methods), which further supports the specificity of our results. Nevertheless, one could have expected that this loop should also have been activated during incorrect meaning attribution (false alarms in the M- condition), as participants reported meaning acquisition. However, the results of Exp. 3 (in which subjects also provided confidence ratings, see Figure 7, Material and methods and Appendix 1), revealed that confidence ratings were significantly lower for false alarms in the M- incorrect condition than for correct responses in the M+ (see below for a similar pattern of results for pleasantness ratings and electrodermal activity). Together, these results strongly suggest that the observed effects were not driven by novelty, attention, task-difficulty, exertion of effort or accuracy, but rather by successful meaning extraction: the VS, HP, and SN/VTA were only engaged when participants correctly learned the meaning of the new-word.

## Is the SN/VTA-HP loop instrumental in enhancing memory formation?

In Exp.1, approximately thirty minutes after the word-learning experiment was completed, all M+ and M- new-words were tested again in a surprise memory test outside the scanner (mean time between encoding and testing: 54 min; range: 37–72 min). Participants still recognized the correct meaning of 68 ± 15% of M+ new-words learned during the test inside the scanner. Participants correctly ascribed no meaning to 68 ± 23% of M- new-words that had been correctly rejected during the prior test. For a second analysis testing the strength of memory formation, M+ correct trials were divided into those in which subjects learned the new-word inside the scanner and still remembered it in the test after the encoding session (*remembered* condition) and those in which the new-word was not correctly identified in the post-scan test (*forgotten* condition; for a similar approach, see *Gruber et al., 2014*). For the analysis of the M+ condition, three subjects with less than 3 forgotten trials had to be excluded. Thus, data from thirty-three subjects were used (average number of

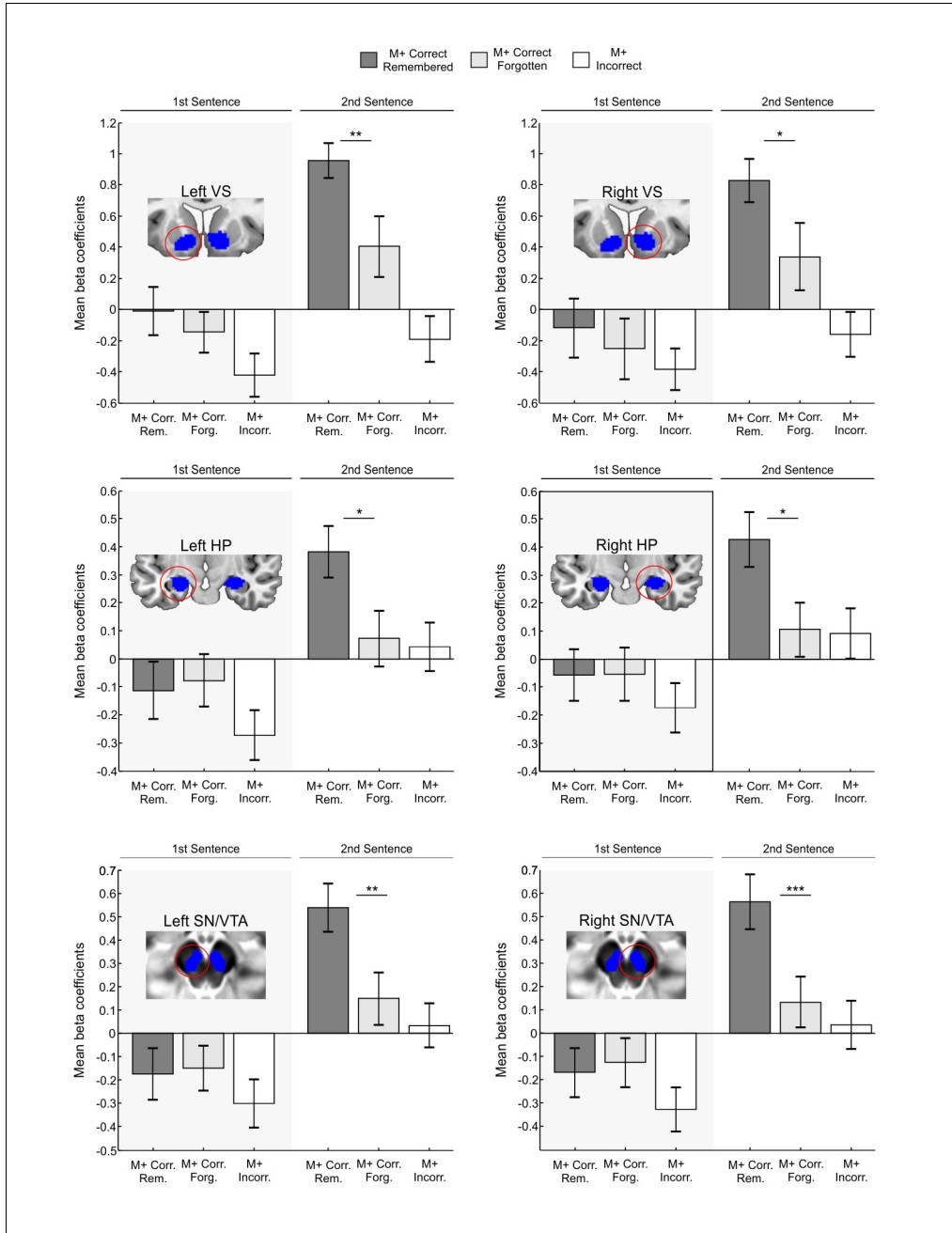

**Figure 3.** ROI analysis of memory effects. Blue areas depict independent ROIs used for beta-extraction overlaid on a canonical T1-weighted template (VS,HP) or the mean proton density normalized template from all subjects (SN/VTA). Bar graphs show mean beta coefficients within ROI for M+ correct trials in which the learned new-word was still remembered during the test performed after the scanning session (remembered) and for M+ correct trials in which the new-word was not properly recognized in the post-scan test (forgotten) with standard error of the mean (dark grey for M+ correct remembered; light grey for M+ correct forgotten; the M+ incorrect condition is shown in white as a control). Paired t-tests showed greater fMRI activity within all ROIs for remembered than for forgotten words, only during the second sentence presentation. M+, congruent meaning extraction possible; VS, Ventral Striatum; HP, Hippocampus; SN/VTA, Substantia Nigra/ Ventral Tegmental Area. ***p<0.005; **p<0.01; *p<0.05.

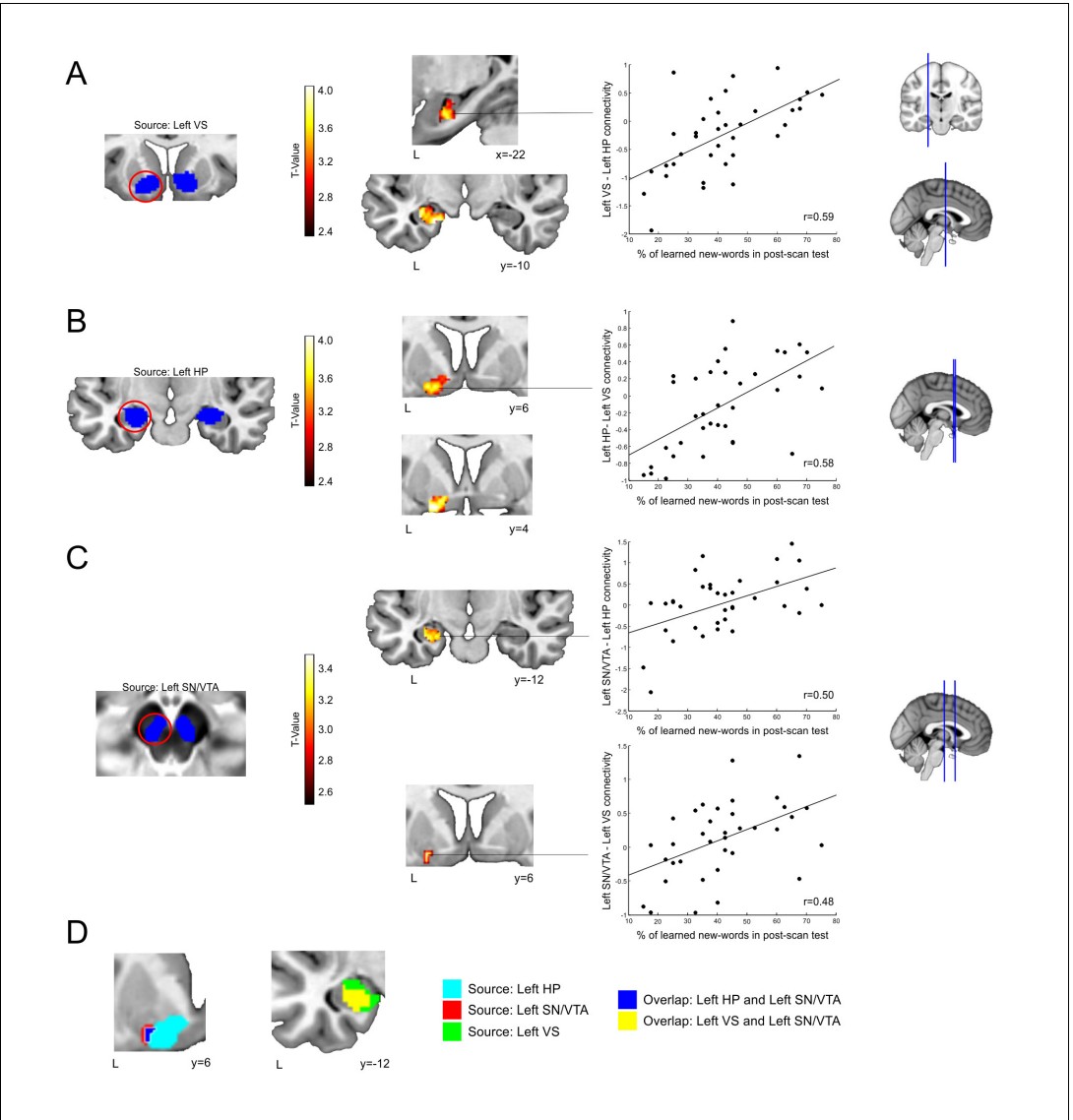

**Figure 4.** Changes in subject-specific connectivity due to individual word-learning success. Blue areas in **A**, **B**, **C** depict the seeds for connectivity-analysis overlaid on a canonical T1-weighted template (VS,HP) or the mean proton density normalized template from all subjects (SN/VTA). Significant correlations between word-learning (learned new-words during the encoding session which were still correctly recognized in the post-scan test) and the change in connectivity in: (**A**) the left VS (source) and the left HP; (**B**) the left HP (source) and the left VS; (**C**) the left SN/VTA (source) and the left HP and the left VS. Results are shown at a p<0.005 uncorrected threshold, with all main peaks within each cluster surviving a p<0.05 FWE corrected threshold within the ROI. The scatter plots display the correlation between the number of learned-words and the mean change in functional connectivity for each particular cluster of interest. Panel (**D**) shows, on the left, the overlap at the left VS (blue) between the correlations with the source at the left SN/VTA (red) and at the left HP (light blue). On the right, the overlap (yellow) between the correlations with the source at the left SN/VTA and at the left VS (green) is shown. Neurological convention is used in both images with MNI coordinates at the bottom left of each slice. L, Left Hemisphere; VS, Ventral Striatum; HP, Hippocampus; SN/VTA, Substantia Nigra/ Ventral Tegmental Area.

trials with standard deviation for each condition: 17.15 ± 6.41 remembered trials and 7.54 ± 3.01 forgotten trials). Beta coefficients were again submitted to a 2 × 2 × 3 × 2 repeated measures ANOVA with the factors Hemisphere (Left, Right), Order (first sentence, second sentence), ROI (VS, HP, SN/VTA), and Memory (Remembered, Forgotten). We found a significant interaction of Order ×

Memory [$F(1,32)$ = 6.398, p<0.017, partial η2 = 0.167] which was not affected by region or hemisphere (all ps>0.812). This interaction reflects greater activation in remembered versus forgotten trials only during the second sentence presentation. Concordantly, subsequent paired t-test comparisons showed significant differences (see *Figure 3*) in remembered versus forgotten trials in the left and right VS, HP, and SN/VTA during the second sentence presentation [left VS, $t(32)$ = 2.78, p<0.009, d = 0.61; right VS, $t(32)$ = 2.40, p<0.022, d = 0.47; left HP, $t(32)$ = 2.60, p<0.014, d = 0.55; right HP, $t(32)$ = 2.58, p<0.015, d = 0.56; left SN/VTA, $t(32)$ = 2.79, p<0.009, d = 0.62; right SN/VTA, $t(32)$ = 3.05, p<0.005, d = 0.65; two-tailed, FDR-corrected at p<0.05; see also Appendix 2 for further supporting analyses in a subset of participants with balanced number of trials in remembered and forgotten conditions]. Importantly, the analysis of remembered vs. forgotten M- trials (eight subjects with fewer than 3 forgotten trials had to be excluded, see Materials and methods; average number of trials with standard deviation for each condition: 17.14 ± 7.03 remembered trials and 7.32 ± 3.24 forgotten trials) revealed no significant interaction of Order × Memory [$F(1,27)$ = 0.004, p>0.948, partial η2 = 0.0001; not affected by region or hemisphere, all ps>0.301] further underlining the specificity of our results to successful meaning extraction (M+ condition). In addition and, as expected, no significant interaction of Order × Memory was found for the control ROI located at the primary visual cortex for M+ or M- trials [$F(1,32)$ = 0.53, p>0.46, partial η2 = 0.017 and $F(1,27)$ = 0.023, p>0.88, partial η2 = 0.001, respectively] which further supports the specificity of our results to reward and memory related regions.

Moreover, if the SN/VTA, HP, and VS form a functional network that promotes memory formation, connectivity among them should also predict word-learning. Thus, we performed a functional connectivity analysis (*psychophysiological interaction*, PPI; *Friston et al., 1997*) which focused on enhanced inter-regional coupling during the congruent condition, when meaning is successfully extracted and remembered in the post-scan test (M+ remembered second sentence) versus when it is not learned (M+ incorrect second sentence).

Subject-specific word-learning success was positively correlated (p-values for peaks are provided using within-ROI family-wise error correction) with the change in connectivity between the **left** VS (source) and the left HP [see *Figure 4A*: 168 voxels; maximum at x = −20, y = −10, z = −22; $t(34)$ = 4.44; p<0.005, d = 1.48]. The correlation of the left VS with the SN/VTA, however, did not survive a correction for multiple comparisons [left SN/VTA, maximum at x = −2, y = −16, z = −14, $t(34)$ = 1.73, p=0.342, d = 0.58; right SN/VTA, maximum at x = 6, y = −16, z = −16, $t(34)$ = 2.58, p=0.098, d = 0.86]. Word-learning was also positively correlated with the change in functional connectivity (see *Figure 4B*) between the left HP (source) and the left VS [100 voxels, maximum at x = V16, y = 2, z = −12, $t(34)$ = 4.34, p<0.004, d = 1.45]. Again, the correlation with the SN/VTA did not survive a correction for multiple comparisons [left SN/VTA, maxima at x = −10, y = −20, z = −10, $t(34)$ = 2.19, p=0.176, d = 0.73; right SN/VTA maxima at x = 10, y = −14, z = −12, $t(34)$ = 2.47, p=0.112, d = 0.82]. Finally, subject-specific word-learning was positively correlated with the change in functional connectivity (see *Figure 4C*) between the left SN/VTA (source) and the left HP [64 voxels, maximum at x = −30, y = −10, z = −20, $t(34)$ = 3.56, p<0.037, d = 1.19] and also with the left VS [10 voxels, maximum at x = −24, y = −8, z = −10, $t(34)$ = 3.33, p<0.04, d = 1.11]. Moreover, there was an overlap at the left VS between the analysis in which the seed was the left HP and that in which the left SN/VTA was used as a source. The same happened at the HP when using the left SN/VTA or the left VS as seeds (see *Figure 4D*). In contrast, analyses using seeds in the **right** SN/VTA, HP, or VS revealed no significant correlations with individual word-learning in any area of interest. In addition, no significant correlations were found at the control ROI located at the primary visual cortex for any of the seeds. Finally, we computed several additional PPI analyses in order to better characterize the nature of the reported effects. First, we repeated the main PPI calculations but focusing on enhanced inter-regional coupling driven by main effects of condition (all M+ versus all M- trials during the second sentence presentation). As expected, no significant correlations with the number of remembered words were found. In the same manner, no effects were found if the PPI analyses focused on M- remembered vs. M- incorrect trials, which further supports the specificity of the connectivity results to the M+ remembered condition. Together, the PPI results from left-sided seeds demonstrate that subject-specific longer lasting word-learning success was related to increased coupling among the VS, HP, and SN/VTA in addition to the intraregional changes in activity within each of these areas in isolation.

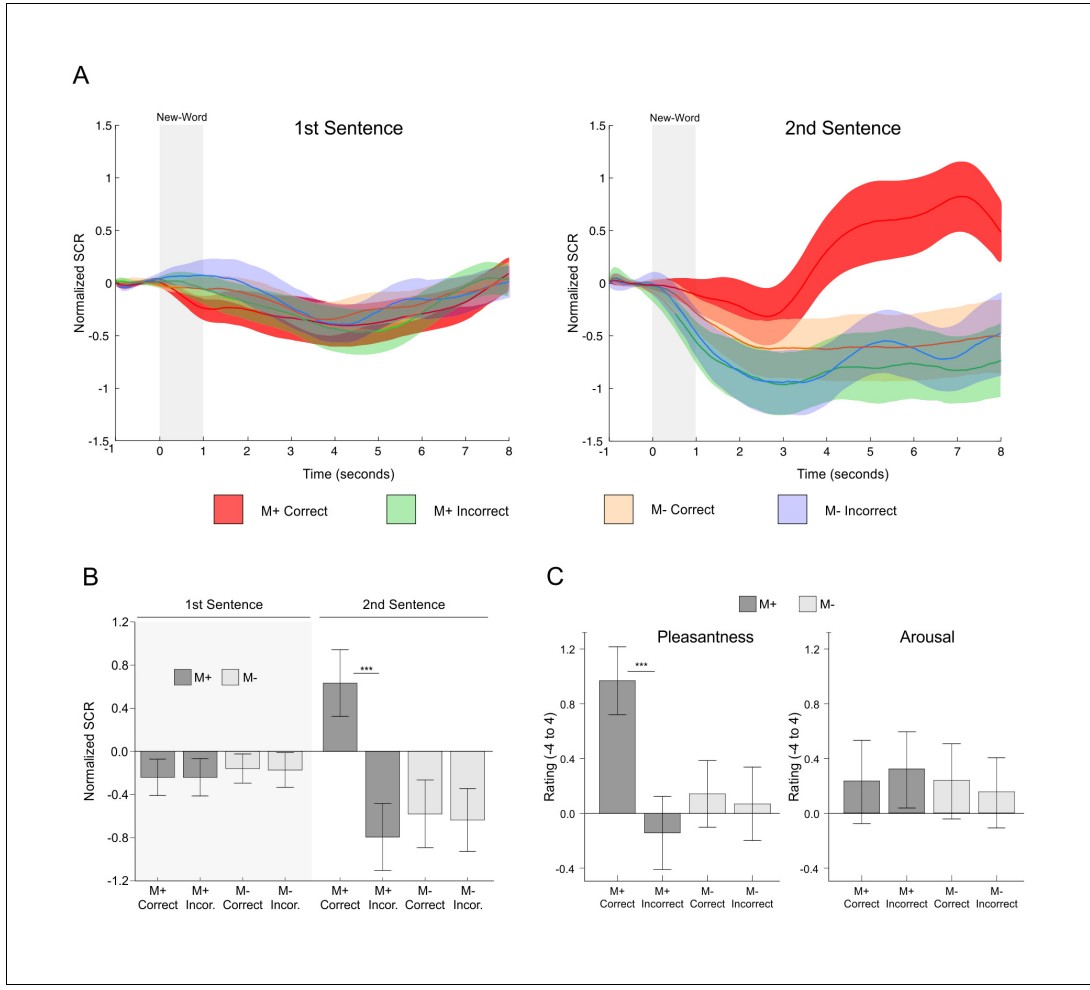

**Figure 5.** SCR signals and pleasantness/arousal scales during the encoding phase of the second experiment. (**A**) Time-course of normalized skin conductance response associated to M+ and M- correct and incorrect conditions during first (left) and second (right) sentence presentation. Solid lines indicate the averaged SCR signal with the corresponding standard error of the mean (red for M+ correct; green for M+ incorrect; light orange for M- correct; light blue for M- incorrect). (**B**) Mean (averaged using the signal form seconds 4 to 8) normalized SCR for each condition of interest with standard error of the mean (dark grey for M+; light grey for M-). Paired t-test comparisons for all correct versus incorrect conditions showed significant differences only for M+ correct versus incorrect trials during second sentence presentation, when participants successfully learned the meaning of a new-word. (**C**) Mean pleasantness/arousal ratings (scale between −4 and 4) for each condition of interest with standard error of the mean (dark grey for M+; light grey for M-). Paired t-test comparisons showed that ratings for pleasantness, but not arousal, were greater for M+ correct trials. See *Figure 7A* for a replication of these effects. M+, congruent meaning extraction possible; M-, congruent meaning extraction impossible. ***p<0.001.

## Does learning success modulate pleasantness and does it promote stable memories after longer intervals?

For the fMRI experiment, the average time interval between encoding of a new-word and its meaning and the presentation of that new-word during the post-scan test was approximately 54 min. If the intrinsic reward modulated LTP, then an enhancement in memory formation should also be evident for longer retention intervals. For this reason, 24 participants completed an additional behavioural experiment (Exp. 2). The experimental design of the fMRI task (Exp. 1) and Exp. 2 were identical except for the three following differences: (i) on-line verbal answers were recorded; (ii) in order to further confirm that our results were reward-related, participants provided subjective arousal and pleasantness ratings using the nine-point visual *Self-Assessment Manikin* scale (SAM;

*Bradley and Lang 1994*); and (iii) subjects completed the retrieval test (which followed the same structure of the fMRI recognition test, see Material and methods; chance level was 33% as three choices were available: no consistent meaning, consistent meaning A, consistent meaning (B) after approximately 24 hr. In addition, SCRs were recorded during the encoding phase as an additional *objective* measure of emotional processing.

In Exp. 2, participants ascribed correct meaning to 62 ± 16% of new-words from the M+ condition during the encoding phase. In 59 ± 16% of the M- trials, participants correctly indicated an absence of coherent meaning. Thus, the pattern of results of experiments 1 and 2 (and also 3, see Appendix 1) was very similar, despite the slight differences in its design: a statistical comparison confirmed that encoding success for all experiments was not different (p>0.91; see Appendix 1).

We then directly tested (as with the fMRI experiment) whether the encoding of a new-word and its meaning modulated SCRs and whether this effect was actually driven by successful meaning extraction or rather by unspecific effects of accuracy, attention, novelty, cognitive load, second word presentation, or experimental condition. Mean normalized SCRs time-courses for all conditions of interest (see *Figure 5A*; one subject was excluded due to problems with data collection) showed an increase in EDA only for M+ correct trials during the second sentence presentation, with a latency of approximately 3 s. Indeed, we found a significant triple interaction of Order × Condition × Response [$F(1,22) = 32.07$, p<0.001, partial $\eta2 = 0.593$; calculated using the mean normalized SCR signal between seconds 4 and 8; the effect is still significant if the mean signal is calculated between seconds 1 to 8, see Appendix 3]. Concordantly, two-tailed paired t-test comparisons for all correct versus incorrect conditions showed significant differences only for M+ correct versus incorrect trials during the second sentence presentation [$t(22) = 4.92$, p<0.001, d = 0.944, FDR-corrected; p>0.6 for all other comparisons; see *Figure 5B*]. These results strongly suggest that the reported effects were not driven by novelty, attention, or cognitive load: SCRs were only enhanced when participants learned the meaning of a new-word.

Turning to the subjective pleasantness and arousal ratings collected during the encoding phase, we found a significant interaction of Condition × Response for pleasantness [$F(1,23) = 15.38$, p<0.001, partial $\eta2 = 0.401$] but not for arousal [$F(1,23) = 0.71$, p>0.4, partial $\eta2 = 0.030$, see *Figure 5C*]. Two-tailed paired t-test comparisons revealed that pleasantness ratings after correct versus incorrect trials were higher for the M+ [$t(23) = 4.87$, p<0.001, d = 0.867, FDR-corrected] but not for the M- condition [$t(23) = 0.58$, p>0.56, d = 0.063]. Thus, high pleasantness ratings were only associated with successful meaning extraction. Importantly, the subjective ratings cannot be explained by participants' compliance with explicit task demands: when asked about the purpose of the study, most subjects answered that it was to measure the effects of reading load on mood (the explanation given to them during briefing, see Materials and methods). None of them answered that the specific purpose of the study was to assess the effect of reward on learning (or similar). Nevertheless, 23 out of 24 subjects did state that successful meaning extraction was rewarding. These results further show that during our task, successful meaning extraction was associated with increased feelings of reward.

Finally, participants carried out an unexpected recognition test 24.03 ± 3.88 hr after the encoding phase. After this 24-hr retention delay, participants still recognized the correct meaning of 42 ± 15% of learned new-words during the encoding phase, significantly above chance level [$t(23) = 2.80$, p<0.010, d = 0.56; chance level was set at 33%, see Materials and methods]. Note that, as expected, participants only recognized the correct meaning of 12 ± 10% of M+ new-words which were not learned during encoding [significantly below the recognition rate for learned M+ new-words, $t(23) = 8.05$, p<0.001, d = 2.23]. Regarding the incongruent condition, participants correctly indicated that 55 ± 27% of M- new-words identified during the encoding phase had no meaning ascribed in the 24-hr test, significantly above chance level [$t(23) = 3.88$, p<0.001, d = 0.78]. However, the 24-hr recognition rate for M- new-words which were *not correctly identified* during the encoding phase was 48 ± 27%, which is not significantly different from the 24-hr recognition rate for M- new-words *correctly identified* during encoding [$t(23) = 1.76$, p>0.090, d = 0.24]. Importantly, in Exp. 3 (which followed the same task structure as Exp. 2, see Materials and methods) participants still recognized 40 ± 15% of learned new-words after a one week retention period [seven days and 4.97 ± 15 hr; $t(22) = 5.16$, p<0.001, d = 1.03], while the percentage of correctly rejected M- words was below chance level after this period of time (16.49 ± 17%).

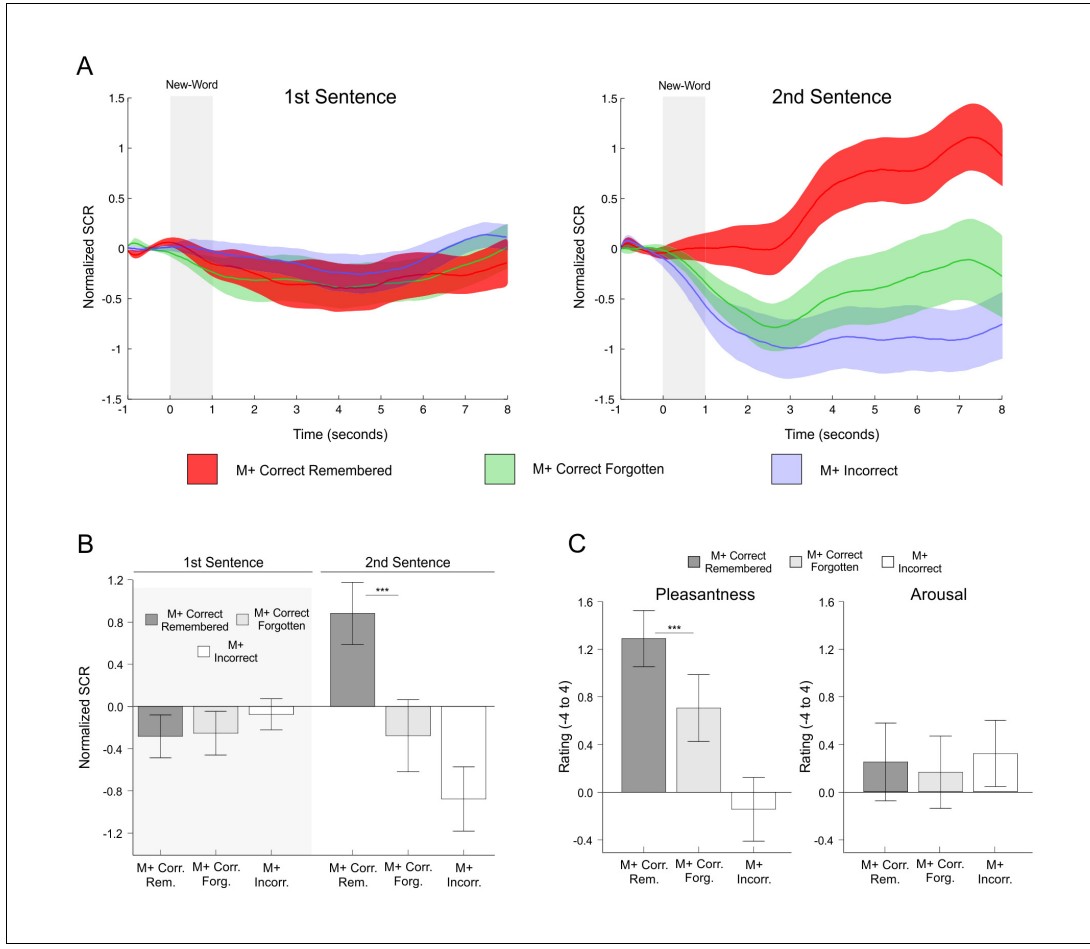

**Figure 6.** SCR signals and pleasantness/arousal scales in the second experiment for remembered new-words after a 24-hr retention delay. (**A**) Time-course of normalized skin conductance response associated to M+ new-words that were remembered or forgotten in the 24 hr test during the first (left) and second (right) sentence presentation. The M+ incorrect conditions are shown as control. Solid lines indicate the averaged SCR signal with the corresponding standard error of the mean (red for M+ remembered; green for M+ forgotten; light blue for M+ incorrect). (**B**) Bar graphs depict mean normalized SCR (averaged 4 to 8 s post-stimulus) for each condition of interest with standard error of the mean (dark grey for M+ remembered; light grey for M+ forgotten). The M+ incorrect condition is shown in white as a control. Paired t-test comparisons for remembered versus forgotten conditions showed significant differences only for M+ words during second sentence presentation. (**C**) Mean pleasantness/arousal ratings (scale between −4 and 4) for M+ correct trials in which the new-word was still remembered during the test performed 24-hr after the encoding session (remembered, dark grey) and for those in which the new-word was forgotten (forgotten, light grey). The M+ incorrect condition is shown in white as a control. Paired t-tests revealed significant differences between remembered and forgotten items for pleasantness, but not for arousal. See **Figure 7B** for a replication of these effects. M+, congruent meaning extraction possible; M-, congruent meaning extraction impossible. ***p<0.005.

We then tested whether enhanced EDA was related to long-term memory effects (i.e., enhanced SCRs for remembered vs. forgotten M+ new-words in the 24-hr test, only during the second sentence presentation). For this analysis, two subjects with fewer than 3 remembered trials were excluded. Mean normalized SCRs time-courses (see **Figure 6A**) showed enhanced EDA for remembered M+ correct trials during second sentence presentation (M+ incorrect trials are also shown as a control). We found a significant interaction of Order × Memory [$F(1,20)$ = 11.57, $p<0.003$, partial $\eta2$ = 0.367, calculated using the mean normalized SCR signal between seconds 4 and 8; the effect is still significant if the mean signal is calculated between seconds 1 to 8, see Appendix 3). Statistical comparisons for all correct versus incorrect conditions showed significant differences for M+

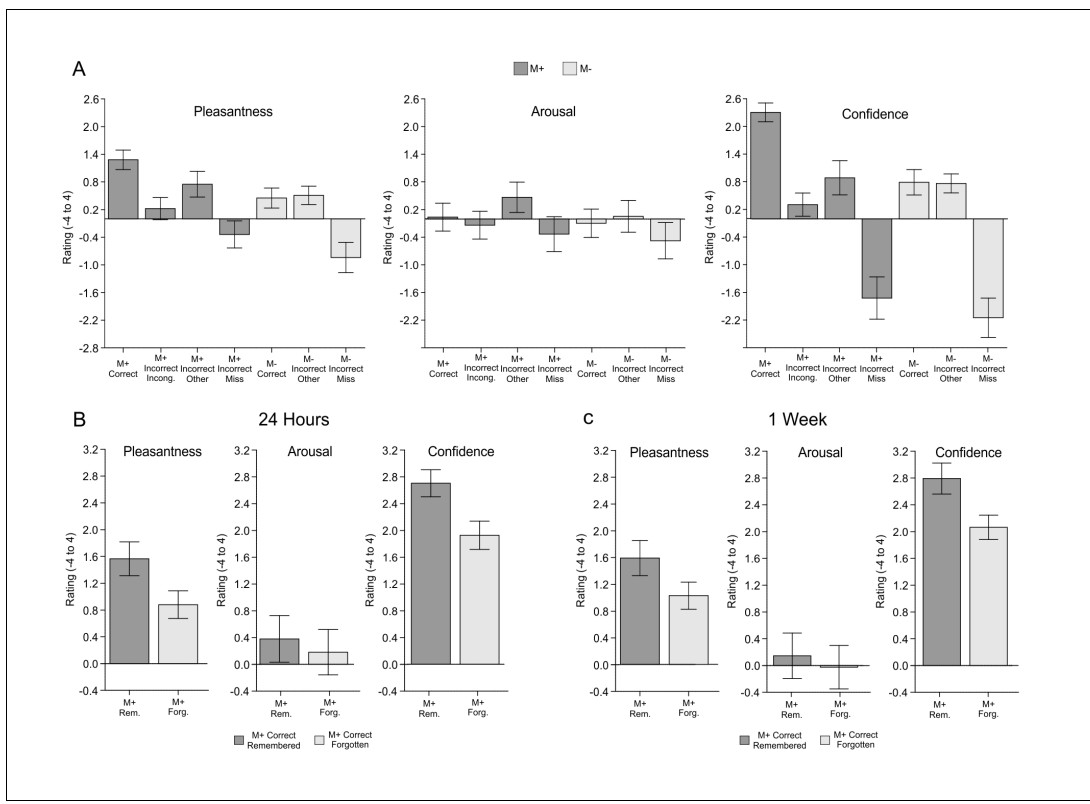

**Figure 7.** Exp. 3. (**A**) Mean pleasantness/arousal/confidence ratings (scale between −4 and 4) for each condition of interest with standard error of the mean (dark grey for M+; light grey for M-). M+ incorrect trials have been divided into incorrect-incongruent (subjects say *Incongruent* instead of providing a meaning), incorrect-other (subjects provide a wrong meaning) and misses (no answer). M- incorrect trials have been divided into incorrect-other (false alarms: subjects provide a meaning for an incongruent new-word) and misses (no answer). (**B**) Exp.3: Mean pleasantness/arousal/confidence ratings (scale between −4 and 4) for M+ correct trials in which the new-word was still remembered during the test performed 24-hr after the encoding session (remembered, dark grey) and for those in which the new-word was forgotten (forgotten, light grey). (**C**) Exp.3: Mean pleasantness/arousal/confidence ratings (scale between −4 and 4) for M+ correct trials in which the new-word was still remembered during the test performed *one week* after the encoding session (remembered, dark grey) and for those in which the new-word was forgotten (forgotten, light grey).M+, congruent meaning extraction possible; M-, congruent meaning extraction impossible.

remembered versus forgotten trials during second [*t*(20) = 3.45, p<0.003, d = 0.78, FDR-corrected] but not during first sentence presentation [*t*(20) = −0.12, p>0.9, d = −0.031; see *Figure 6B*]. As expected, when computing the same analysis for M- trials (five subjects with fewer than threeremembered or forgotten trials were excluded), no significant interaction of Order × Memory was found when using the mean normalized SCR signal between seconds 4 and 8 [F(1,17) = 0.01, p>0.915, partial $\eta2$ = 0.001] or between seconds 1 to 8 [the whole time-course, F(1,17) = 0.05, p>0.818, partial $\eta2$ = 0.003]. These results indicate that EDA signals were not modulated by memory effects related to the M- condition. Together, the EDA results indicate that remembered new-words elicited enhanced SCRs during the encoding phase as compared to forgotten ones.

Concordantly, subjective pleasantness ratings were also higher for remembered than for forgotten M+ new-words in the 24-hr recognition test [*t*(23) = 4.30, p<0.001, d = 0.454; see *Figure 6C*], while no difference in arousal ratings was found [*t*(23) = 0.68, p>0.25, d = 0.055]. These effects were replicated in Exp. 3 for a 1-week retention period (see Appendix 1 and *Figure 7*). Once again, and, as expected, the same analysis for the M- condition (three subjects were excluded from this analysis as they did not correctly identify in the 24-hr test any of the M- new-words correctly rejected during the encoding phase) showed no difference in subjective pleasantness [t(20) = −1.44, p>0.16,

d = −0.12] or arousal ratings [t(20) = −0.33, p>0.73, d = −0.039] for M- remembered vs. forgotten trials. Thus, neither pleasantness nor arousal scales were modulated by memory effects related to the M- condition. All in all, these results suggest that intrinsic reward, derived from an internal evaluation of learning success for the M+ condition only, had a modulatory effect on long-term memory.

## Discussion

The goal of the present study was to test whether an intrinsic signal, triggered by successful internally-guided learning, could enhance memory formation through the activation of a network formed by the VS, the HP, and the SN/VTA. The fMRI-results demonstrate that successful meaning extraction for a new-word, in the absence of any external feedback, enhanced fMRI signals within the entire SN/VTA-VS-HP loop, with the observed activity not being caused by novelty, attention, task-difficulty, or exertion of effort, but rather by reward-related effects (*Figure 2*). Moreover, in a second experiment, objective physiological measures (EDA) and subjective pleasantness ratings were only enhanced during successful meaning extraction (*Figure 5*), which further demonstrates that learning was associated with increased reward processing. Regarding our main hypothesis—the effect of intrinsic reward on memory—new-words learned and later remembered elicited greater fMRI activity and functional connectivity within the VS, HP, and SN/VTA than those that were forgotten (*Figures 3* and *4*). Finally, and most importantly, Exp. 2 also showed that remembered new-words after a 24-hr retention delay (and after seven days in Exp. 3) were closely tied to enhanced SCRs and increased ratings of pleasantness during encoding, suggesting that intrinsic reward—driven by self-monitoring of correct performance—did have an effect on memory formation (*Figures 6* and *7*). This study extends our previous results (*Ripollés et al., 2014*) by showing that successful learning, in the absence of external feedback or reward, engages a complex subcortical network—that includes not only reward, but also memory and dopamine related regions—which seems to modulate the entrance of new information into long-term memory. All in all, this is the first study, to the best of our knowledge, to provide a neural mechanism—the SN/VTA-HP loop—which might subserve reward-related memory enhancements when the *reward is intrinsic and triggered by an internal evaluation of correct performance*. Thus, learning, under certain circumstances, could be fuelling itself through intrinsic reward-related processes. Remarkably, this engagement was only observed for the meaningful condition and not during incorrect meaning attribution in the incongruent condition (M-incorrect). Likewise, neither subjective pleasantness ratings nor EDA measures were affected by M-incorrect trials, suggesting that the incorrect behavioural responses are somehow corrected by the internal monitoring system and thus did not trigger learning (see confidence ratings for Exp. 3 in Appendix 1 and *Figure 7*).

Our results are in line with those of *Kizilirmak et al. (2015)* who showed that, behaviourally, successful generation of a solution to a problem in the absence of explicit feedback was related to both positive affect and enhanced memory formation. Importantly, we extend those findings to meaningful learning of new-words and show that objective measures of emotional responses (EDA) are affected as well. In this vein, our results are in accord with accumulating evidence suggesting that emotions can influence decision making and learning (*Dunsmoor et al., 2015*; *Adcock et al., 2006*; *Kizilirmak et al., 2015*; *Bechara and Damasio, 2005*). In addition, the intrinsic nature of the reward signals triggered by successful learning in the present experiments finds parallels in reinforcement learning models which show that learning systems incorporating intrinsic (in addition to extrinsic) reward signals surpass those based only on extrinsic ones (*Schultz, 2015*; *Barto et al., 2004*). Crucially, in our paradigm, internally generated signals were also fundamental: fMRI activity within the SN/VTA, HP and VS was only modulated by successful learning (i.e., when participants realized that they had extracted a correct meaning; also, confidence and pleasantness ratings were only high during correct trials, see *Figure 7*).

Importantly, there is accruing evidence suggesting that the dopaminergic release at the HP is fundamental to promote stable memories (*Bethus et al., 2010*; *Frey et al., 1990*; *Hansen and Manahan-Vaughan, 2014*; *Huang and Kandel, 1995*; *McNamara et al., 2014*; *Rossato et al., 2009*). Indeed, several studies have used stimuli that trigger the release of dopamine—especially reward—to induce memory enhancements in human adults (i.e., activating the SN/VTA by a rewarding stimulus such as money enhances the memory for events present during, after and even before the reward was delivered; *Shohamy and Adcock, 2010*; *Krebs et al., 2009*; *Wittman et al., 2005*,

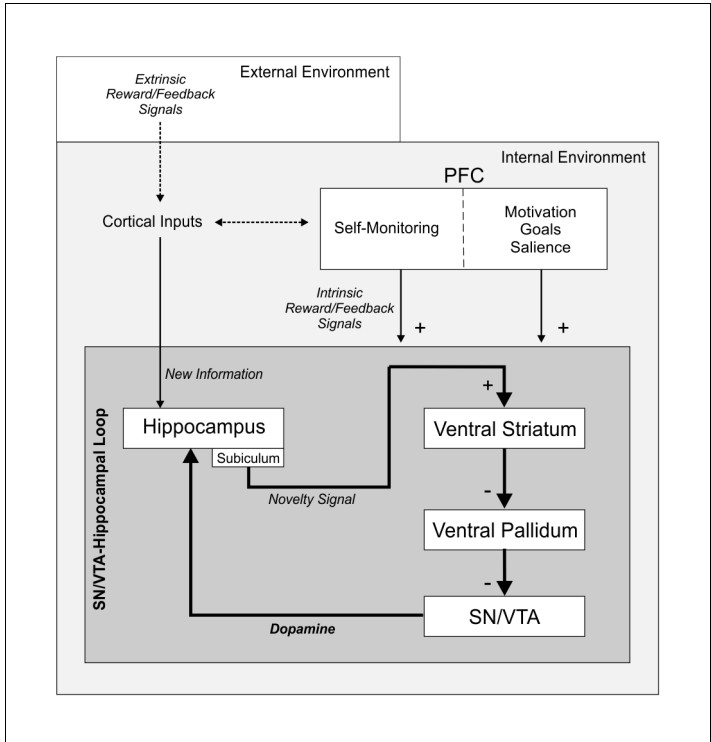

**Figure 8.** The SN/VTA-Hippocampal loop. In the downward arm of the loop, activation starts when new information that needs to be stored in memory 'arrives' at the HP via cortical inputs. A signal is then sent to the SN/VTA through the subiculum of the hippocampus, the VS, and the ventral pallidum. Neurons at the SN/VTA are disinhibited by the arriving signal from the HP, which facilitates their dopaminergic firing. In the upward arm of the loop, dopamine is released back into the hippocampus, which in turn enhances memory formation and learning through long term potentiation processes. We suggest that—in the same manner as an extrinsic reward modulates the HP, VS, and SN/VTA, and promotes memory benefits—the activity within the SN/VTA-HP loop can be induced by an intrinsic reward inherent to the process of learning itself and triggered by an internal monitoring of correct performance; and that this intrinsic reward ultimately promotes the storage of new information into long-term memory via dopaminergic modulation of the midbrain. We hypothesize that the prefrontal cortex is fundamental for self-monitoring of correct performance (see *Ripollés et al., 2014* for results showing activity within the inferior, middle and superior frontal gyrus while participants were engaged in the same learning task). PFC, prefrontal cortex.

*2008*; *Adcock et al., 2006*; *Murty and Adcock, 2014*; *Wolosin et al., 2012*). Importantly, not only extrinsic signals but also intrinsic motivational states can enhance memory formation. For example: in a recent study, both the VS and HP showed enhanced activity during the anticipation of trivia answers that were later remembered, only when participants were engaged in states of high curiosity (*Gruber et al., 2014*). Thus, both anticipation of explicit rewards and intrinsic motivational states can promote memory formation, and both engage the SN/VTA, HP and VS. Likewise, a very recent study demonstrated that after neurofeedback training, human adults displayed the ability to sustain the VTA activation without the need for an external reward (*MacInnes et al., 2016*). *MacInnes et al. (2016)* also showed that connectivity between the VTA and the HP was enhanced during and after neurofeedback training (thus relating their results to the SN/VTA-Hippocampal loop), although they did not study the impact of this volitional sustained VTA activation on learning. Another source of evidence linking dopamine and memory enhancements comes from pharmacological studies that tried to rise the levels of dopamine in the human brain. In this regard, research has shown memory benefits after the intake of dexamphetamine, which blocks dopaminergic and adrenergic re-uptake, and levodopa (a dopamine precursor; *Breitenstein et al., 2004*; *Whiting et al., 2007*, *2008*; *Shellshear et al., 2015*; *Knecht et al., 2004*; *Chowdhury et al., 2012*; *Bunzeck et al., 2014*).

To summarize, there is converging evidence that explicit reward (i.e., money), intrinsic motivational states (i.e., curiosity) and even pharmacological manipulation (i.e., levodopa intake) can increase the levels of dopamine at the HP, potentially inducing memory benefits (*Shohamy and Adcock, 2010*; *Gruber et al., 2014*). We suggest that the activity within the SN/VTA-HP loop can also be modulated by an *intrinsic reward*—inherent to the process of learning itself and triggered by internal self-monitoring of correct performance—which ultimately promotes the storage of new information into long-term memory via dopaminergic modulation of the midbrain (see *Figure 8*). Our results do support this working hypothesis, as a activity within all three areas of interest was larger for remembered than for forgotten learned new-words in the fMRI experiment. In this vein, dopamine release at the HP can mediate both early (minutes) and late (hours) LTP, although stronger effects have been reported for the latter (*Otmakhova and Lisman, 1996*; *Bethus et al., 2010*; *Smith et al., 2005*; for a review see *Lisman et al., 2011*). The results from Exp. 2—in which pleasantness ratings and SCRs were higher for new-words that were remembered after a 24-hr retention delay—are crucial to support this notion, suggesting that the emotional impact associated to successful learning had an effect on long-term memory formation (see also Exp. 3 for a replication of the behavioural effects after a 1-week retention period). Note that the new-words to be learned were all novel and that the meanings associated with them were equally familiar. Therefore, it is unlikely that subjects may have learned some new-words over others based on their specific characteristics. In addition, the SN/VTA-HP loop was not activated by M- new-words, suggesting that in our paradigm this loop was only triggered by meaningful learning. However and, in spite of all this evidence and although fMRI signals within the SN/VTA have been linked to dopaminergic signalling (*Düzel et al., 2009*; *Schott et al., 2008*; *Knutson et al., 2007*; *Salimpoor et al., 2011*; *Ferenczi et al., 2016*), enhanced fMRI activity in isolation only provides indirect evidence for dopamine release. Nevertheless, from a comparative neurobiological perspective, our results converge with previous animal findings revealing that midbrain dopaminergic neurons signal not only primary but also cognitive rewards (*Bromberg-Martin et al., 2009*) and with a recent study showing that midbrain-dopaminergic neurons of songbirds contribute to song learning, which is also mediated by an internal evaluation of correct performance (i.e., songbirds learn to sing by comparing their song to their memory trace of an adult's song; *Mandelblat-Cerf et al., 2014*).

Importantly, functional connectivity results further corroborated our interpretation, as they showed that the connection strength among the HP, VS, and SN/VTA predicted subject-specific word-learning success for remembered words in the post-scan test (see *Figure 4*). These connectivity results are of utmost importance, as they reveal that the coupling among the HP, VS, and SN/VTA predicted subject-specific memory enhancements (see *Adcock et al., 2006*; *Wolosin et al., 2012* for similar effects of external reward on SN/VTA-HP connectivity). Regarding the connectivity results, an important limitation of the current work is that, given the poor temporal resolution of fMRI, we cannot fully address the temporal sequence of events during successful word learning. In principle, effective connectivity analyses could have been computed in order to assess the directionality of the reported effects. However, we acquired fMRI volumes in an interleaved order, which is suboptimal for Dynamic Causal Modeling and other types of effective connectivity analyses (*Stephan et al., 2010*). Future research could take advantage of concurrent EEG and fMRI set-ups and a continuous acquisition of fMRI data to further assess the relationships among the different regions of the SN/VTA-HP loop. In addition, future studies could also try to directly measure the release of dopamine at the HP during learning by means of Positron Emission Tomography (PET) or by using pharmacological interventions (e.g., levodopa intake).

Although our results (coming from fMRI data, EDA signals and subjective pleasantness ratings) suggest that the SN/VTA-HP loop was engaged by an intrinsic reward-related signal that was associated with the participants' monitoring of their learning success, recent research proposes several alternative interpretations on how this circuit could be engaged in the absence of feedback. On the one hand, the simple act of choosing (i.e., participants had perceived control over their learning) can induce enhancements in memory, with the latter being associated with enhanced connectivity between the striatum and the HP (*Murty et al., 2015*). Importantly, Murty and colleagues emphasized that, in their task, participants were active learners that had a perceived sense of agency over their environment. In a similar manner, in our paradigm participants managed to actively guide their own learning and to successfully monitor their performance, and this also had an effect on memory success (see *Figure 7*). However, participants were asked to choose in both M+ and M- conditions,

making decisions in both circumstances. Therefore, differences in choice-related activity should have less influence in our paradigm. On the other hand, *Duncan et al. (2014)* showed that connectivity between the HP and the SN/VTA during encoding of a classical associative task, in the absence of feedback, was predictive of long-term memory success, providing further support for the SN/VTA-HP loop. It is important to emphasize that according to the SN/VTA-HP model (*Lisman and Grace, 2005*) this network will be activated when the HP detects that novel information has to be encoded (i.e., by a novelty signal). Although the temporal resolution of fMRI data prevents us from fully disentangling the origin of the detected activity within the SN/VTA-HP loop, given that the ROIs were selected using a meta-analysis on reward and taking into account that the M- and NR condition serve as a control for novelty effects, we suggest that the reported effects are related to intrinsic reward-related (rather than to novelty-related) signals that increased activity within the loop and ultimately enhanced learning. In addition, subjective goals can also modulate activity within the striatum (*Han et al., 2010*) and, therefore, in our paradigm, participants' internal motivation towards correctly completing the task could also have modulated activity within the areas of the loop. Note, however, that the incongruent condition (M-) partially controls for this possibility, as participants were equally prompted to correctly complete the M+ and M- conditions and in both cases a successful solution could be achieved.

Moreover, research in human adults has shown that reward and dopamine related regions can use internally generated signals of self-performance to guide perceptual learning in the absence of external feedback (see for a review, *Daniel and Pollmann, 2014*). In particular, in two recent studies, activity within the VS (*Daniel and Pollmann, 2012*) and within the VS and SN/VTA (*Guggenmos et al., 2016*) was related to the generation of a reward/confidence prediction error (i. e., activity was induced if participants thought that they performed better than they had expected in a perceptual task). In *Daniel and Pollmann (2012)*, *Guggenmos et al. (2016)* and in the present work, learning occurs without external feedback, is driven by mesolimbic areas and internally generated signals of correct performance play a fundamental role. However the output to be learned and the mechanism subserving this learning could still be different: while the results of the two aforementioned studies are based on the well-known role of dopamine in the codification of a reward prediction error (*Schultz, 1998*), we suggest that our learning is driven by the effects that dopamine has in LTP when it is released at the HP (*Shohamy and Adcock, 2010*).

Finally, our results also have specific implications for language learning. The fact that the SN/VTA-HP loop can mediate the entrance of new-words into memory fits well with current perspectives which suggest that language requires the convergence of multiple neural functions—many shared with other species—such as memory, attention, and, crucial to our hypothesis, reward-related mechanisms (*Fitch, 2010*; *Ripollés et al., 2014*). In this vein, several studies have linked the intake of levodopa with an enhancement in word-learning (*Knecht et al., 2004*; *Shellshear et al., 2015*). By showing that the coupling between the SN/VTA, VS, and HP can modulate successful word-learning from context—a natural learning process that is thought to be related to vocabulary growth during our lifespan (*Nagy et al., 1985*) and that occurs without the need of explicit reward, feedback or external guidance—we provide a mechanistic explanation on how dopamine could enhance certain forms of second language acquisition. We suggest that language models should be extended to include the reward-memory SN/VTA-HP loop, as it might partially subserve specific word-learning processes (see *Figure 8*; *Davis and Gaskell, 2009*; *Rodríguez-Fornells et al., 2009*).

In conclusion, we show that an intrinsic signal, triggered by successful learning, can modulate the entrance of new information into memory through the activation of the entire SN/VTA-VS-HP loop. We propose that this signal is reward-related and that dopamine release plays a crucial role in this process. Finally, the fact that word-learning can fuel itself through reward-related mechanisms that have major implications for second language acquisition and, most importantly, for the recovery from neurological disorders such as aphasia.

## Materials and methods

For significant interactions partial eta squares ($\eta2$) is provided as a measure of effect size. For significant differences measured with t-tests, Cohen's d is provided after applying Hedges' correction (the average of the standard deviation of the variables being compared was used as a standardizer; *Cumming, 2012*).

## Participants

Forty German speakers were recruited from the student population at Otto-von-Guericke University (Magdeburg, Germany) for fMRI-Exp. 1. Four participants were rejected due to excessive head movements during the MRI session (abrupt head motion exceeding 4 mm). For Exps. 2 and 3, twenty-four different Spanish speakers were recruited from the student population at the Universitat de Barcelona (Barcelona, Spain). All participants were right handed, gave written consent, and received compensation for their participation in accord with local ethics. Thus, the final group consisted of thirty-six participants (mean age, SD = 24.75 ± 4.7, 17 women) for Exp. 1 (same group as in *Ripollés et al., 2014*), 24 subjects (25.16± 3.73 years, 15 women) for Exp. 2 and 24 for Exp. 3 (21.16 ± 3.27 years, 18 women). For fMRI Exp. 1, the sample size was chosen based on the recommendation that, in order to achieve 80% of power, at least 30 participants should be included in an experiment in which the expected effect size is medium to large (*Cohen, 1988*). We decided to recruit 40 participants anticipating possible problems (e.g., participant exclusion due to excessive movement). To determine sample sizes for Experiments 2 and 3 we took into account behavioural results from Exp. 1, in which participants still remembered 68% (*SD* = 15) of learned new-words in the test carried out approximately 30 min after the encoding session ended (chance level was 33%; effect size of Cohen's d = 2.13). Since testing occurred after a 24-hr retention delay, we assumed a more conservative (but still large) effect size of 0.8 (*Cohen, 1988*). A sample size analysis, calculated using the G*Power program, showed that a sample size of 12 was required to ensure 80% of power to detect a significant effect of encoding at the 5% significance level. We decided to double the calculated a priori sample size, as we expected the 24-hr retention delay to lower the recognition success as compared to Exp. 1.

## Word-learning task for fMRI exp. 1

### Experimental task

Stimuli were presented using the Psychophysics Toolbox 3.09 (*Brainard, 1997*) and Matlab version R2011b (7.13.0.564, 32 bit). Stimuli consisted of 80 pairs of 7 word-long German sentences ending in a new-word that stood for a noun (mean frequency 46.5 per million; *Mestres-Missé et al., 2010*). The two sentences for each noun were built with an increasing degree of contextual constraint (*Mestres-Missé et al., 2014*). The new-words respected the phonotactic rules of German and were built by changing one or two letters of an existing word. In the current experiment, only half of the pairs of sentences disambiguated multiple meanings, thus enabling the extraction of a correct meaning for the new-word (M+ condition; e.g., 1. ''Every Sunday the grandmother went to the *jedin*'' 2. ''The man was buried in the *jedin*''; *jedin* means graveyard and is congruent with both the first and second sentence; see *Figure 1A*). For the other 40 pairs, second sentences were scrambled so that they no longer matched their original first sentence. In this case, the new-word was not associated with a congruent meaning across the sentences (M- condition; e.g., 1. ''Every night the astronomer watched the *heutil*''. Moon is one possible meaning of *heutil*. 2. ''In the morning break co-workers drink *heutil*.'' Coffee is now one of the possible meanings of *heutil*, which is not congruent with the first sentence; see *Figure 1A*). These constituted the M- condition in which congruent meaning extraction was not possible. To ensure that both stimulus types were equally comparable, participants were told that it was just as crucial to learn the words of the M+ condition as it was to correctly reject the new-words from the M- condition. In this manner, participants were able to use contextual information to self-monitor their correct performance. In addition, non-readable sentences (NR; see *Figure 1A*) created from the M+ and M- stimuli by converting each letter into a symbol were also presented as a control. No motor responses were required during the learning runs.

Four pairs of M+, four pairs of M-, and two pairs of NR conditions were presented per fMRI run (10 runs total). Therefore, a total of 40 new-words from the M+ and 40 from the M- conditions were presented during the whole experiment. In order to achieve an ecologically valid paradigm, presentation of the first and second sentences with the same new-word at the end were separated in time. The 4 first sentences of each of the M+ and M- conditions (a total of eight new-words) plus 2 'sentences' of the NR condition were presented in a pseudo-randomized order (e.g., M+1A, M-1A, M-1B, NR1A, M-1C, M+1B, M+1C, NR1B, M+1D, M-1D). Then, the second 'pair' sentences of both M+ and M- conditions were presented (i.e., second presentation of the identical eight new-words), again in a pseudo-randomized order including 2 'sentences' of the NR condition (e.g., M-2C, M-2B,

NR2A, M+2B, M+2D, M-2D, M+2C, M+2A, M-2A, NR2B). The temporal order of the different new-words during the first sentence presentation was not related in any systematic way to the order of presentation of the same new-words for their second sentence.

As the scanning set-up did not allow for online recording of responses, immediately after each encoding run, participants had to complete a test that was devised to assess their performance (i.e., to assess the words that they had correctly encoded or that they had correctly rejected). On average, mean time between the second presentation of a new word during the encoding block and the presentation of that very same word during testing was approximately 45 s. Participants were presented with a new-word at the centre of the screen with two possible meanings below: one on the left and one on the right. In each test, all 4 M+ and 4 M- new-words presented during a encoding run were tested in a pseudo-randomized order. If the new-word tested did not have a congruent meaning associated between the first and the second sentence, and thus correct meaning extraction was not possible (M- condition), participants had to press a button located in their left hand. In this case, the two possible meanings presented served as foils: one was the meaning evoked by the second sentence of the M- new-word being tested; the other word shown was the meaning evoked by another second sentence presented in the same run as the new-word being tested. Instead, if the new-word tested had a consistent meaning through the first and second sentence, and thus correct meaning extraction was possible (M+ condition), participants had to select the correct meaning using a two-button pad placed on their right hand. In this case, one of the two possible meanings was correct and the other, which served as a foil, was the meaning of another new-word presented in the same run. Therefore, chance level was at 33% as, for both the M+ and M- conditions, three response options were available (no consistent meaning, consistent meaning presented on the left of the screen, consistent meaning presented on the right of the screen).

Approximately 30–35 min after the word-learning task ended and once outside the scanner, participants had to complete a surprise recognition test. In this post-scan test, participants were presented with all the 40 M+ and 40 M- new-words used in the experiment. They were instructed to proceed exactly as in the previous test. The only difference was that the pairings between true meanings and foils were different than those tested inside the scanner. In this post-scan test, mean time between encoding (inside the scanner) and testing (after the fMRI session) of a particular new-word was 54 min. Therefore, this test was devised in order to assess which of the learned words inside the scanner were still remembered and which of them had been forgotten after almost 1 hr of retention period. Participants were aware that they would complete a test after each encoding run and were instructed on how to answer it. It was made explicit that they would assess both M+ and M- new-words during these test phases. However, only after the fMRI task ended were they told that they had to complete the recognition test. All participants completed a training block before entering the scanner in order to familiarize them with the task and the recognition test. See *Figure 1B* for timing details of each trial.

## Scanning parameters

MRI data were collected on a 3T scanner (Siemens Magnetom Trio) using an eight-channel phased-array coil (Siemens, Erlangen, Germany). The session started with the acquisition of an inversion recovery prepared echo-planar imaging sequence (IR-EPI; TR = 15000 ms, TE = 21 ms, TI = 1450 ms, flip angle = 90°, slice thickness = 3.8 mm, 3 mm in plane resolution, 34 slices, matrix size = 80 × 80) in order to allow precise anatomical co-registration with functional data. After this, 10 runs of 92 sequential whole-brain volumes of EPI images sensitive to blood-oxygenation level-dependent contrast (Gradient Echo EPI; TR = 2000 ms, TE = 30 ms, flip angle = 80°, slice thickness = 3.8 mm, 3 mm in plane resolution, 34 slices, matrix size = 80 × 80; interleaved acquisition) were acquired for the word-learning task. Finally, a proton density (PD; TR = 8100 ms, TE = 15 ms, flip angle = 150°, slice thickness = 2.09 mm, 0.9375 mm in plane resolution, 68 slices, matrix size = 256 × 256) structural image was also acquired, in order to properly identify the SN/VTA (*Boehler et al., 2011*; *Oikawa et al., 2002*).

## ROI creation

Given our explicit a-priori hypothesis regarding the VS, HP, and SN/VTA, an ROI analysis was performed. In order to avoid circularity (*Kriegeskorte et al., 2009*), the ROIs were created using

independent data. The VS ROI was created using the results from an independent monetary gambling task from a previous experiment dealing with the same subjects (contrast: gains>losses, thresholded at FWE-corrected p<0.001; *Ripollés et al., 2014*). The HP ROI was created using the toolbox Wfu-pickatlas (*Maldjian et al., 2003*, *2004*) and the Automated Anatomical Labelling Atlas (*Tzourio-Mazoyer et al., 2002*). The SN/VTA ROI was created using the individual proton density images (that we acquired specifically for this purpose) from all 36 participants in the following way: these images were first segmented using the New Segment tool from SPM8 (an improved version of the 'Unified Segmentation' algorithm; *Ashburner and Friston, 2005*). The GM and WM tissue probability maps obtained during the segmentation were then fed to DARTEL (*Ashburner, 2007*) to achieve spatial normalization into MNI space. DARTEL normalization alternates between computing an average template of GM segmentation from all subjects and warping all subjects' GM tissue maps into a better alignment with the template created (*Ashburner, 2009*). An MNI proton density group template was then obtained by calculating the mean of all normalized images. The SN/VTA ROI was delineated using this template. The present study did not attempt to separate the SN from the VTA as, in humans, these two structures build a continuous dorsal tier (*Boehler et al., 2011*; *Ahsan et al., 2007*). However, it is important to note that, in humans, most of the dopaminergic cells are located within the SN (*Björklund and Dunnett, 2007*). In order to maximize sensitivity within our ROIs, we performed a meta-analysis using NeuroSynth (a platform for large-scale, automated meta-analysis of fMRI data; www.neurosynth.org; *Yarkoni et al., 2011*). We calculated a term-based search on reward that resulted in 560 studies (search performed on June 30, 2015). Then, a reverse inference mask (which represented the probability that the term reward was associated with a particular activation) was generated. We then refined the three previously created ROIs (VS, HP, and SN/VTA) by masking them with the results of the NeuroSynth meta-analysis. In other words, each final ROI only contained voxels that were part of the original ROIs (created from a functional localizer used in our previous work for the VS; an anatomical atlas for the HP; and proton density images for the SN/VTA) and that were also reward-related according to the meta-analysis. This procedure followed recommendations from *Gruber et al. (2014)*, in which the SN/VTA-HP loop was also studied by means of ROIs. In the case of the VS, the original functional ROI remained intact after the masking with the NeuroSynth analysis (i.e., all the voxels included in the functional localizer were reward-related according to the NeuroSynth meta-analysis; number of voxels in the final ROIs: 334 for left VS, 289 for right VS). Vast areas of the originally SN/VTA ROI (defined using proton density images from all our subjects) were also included after masking with the reward meta-analysis, except for very inferior portions (number of voxels in the final ROIs: 101 for left SN/VTA, 99 for right SN/VTA). The original HP ROIs that were created from an anatomical atlas were restricted to the head and anterior body of the hippocampi after masking with the NeuroSynth meta-analysis (number of voxels in the final ROIs: 450 for left HP, 434 for right HP). Finally, an extra ROI covering the primary visual cortex (BA17) was created using the toolbox Wfu-pickatlas and the Automated Anatomical Labelling Atlas, as with the HP ROI (*Maldjian et al., 2003*, *2004*; *Tzourio-Mazoyer et al., 2002*). This ROI served as control region to further assess the specificity of our findings (number of voxels in the final ROIs: 362 for left BA17, 370 for right BA17). Therefore, all created ROIs were independent of the word-learning paradigm, thus avoiding 'double-dipping'.

## fMRI preprocessing

Data were preprocessed using Statistical Parameter Mapping software (SPM8, Wellcome Trust Centre for Neuroimaging, University College, London, UK, www.fil.ion.ucl.ac.uk/spm/). Functional runs were first realigned and a mean image of all the EPIs was created. The inversion recovery image was co-registered to the mean EPI image and then segmented into grey and white matter (GM; WM) by means of the Unified Segmentation algorithm (*Ashburner and Friston, 2005*). After an initial 12-parameter affine transformation of the GM tissue probability map to the GM Montreal Neurology Institute (MNI) template included with SMP8, the resulting normalization parameters were applied to the whole functional set. Finally, functional EPI volumes were re-sampled into $2 \times 2 \times 2$ mm voxels and spatially smoothed with an 8 mm FWHM kernel.

An event-related design matrix was specified using the canonical hemodynamic response function. Trial onsets were modeled at the moment of the presentation of the new-word. M+ and M- conditions were classified as correct or incorrect using the test performed after each encoding run.

Hence, ten different conditions were specified: M+ correct first sentence, M+ incorrect first sentence, M- correct first sentence, M- incorrect first sentence, NR first sentence, M+ correct second sentence, M+ incorrect second sentence, M- correct second sentence, M- incorrect second sentence, and NR second sentence. Data were high-pass filtered (to a maximum of 1/128 Hz) and serial autocorrelations were estimated using an autoregressive (AR(1)) model. Remaining motion effects were minimized by also including the estimated movement parameters in the model. First-level contrasts were specified for all participants using each condition against the implicit baseline.

## ROI analysis I: controlling for novelty and task difficulty

In order to control for novelty and task-difficulty and ensure that the detected activity was reward-related, a full-factorial ROI analysis was performed. Individual beta coefficients for each participant were extracted from the fMRI analysis of the word-learning task and submitted to a $2 \times 2 \times 3 \times 2 \times 2$ repeated measures analysis of variance (ANOVA) with the factors Hemisphere (Left, Right), Order (first sentence, second sentence), ROI (VS, HP, SN/VTA), Condition (M+, M-), and Response (Correct, Incorrect). For beta extraction, individual beta coefficients (extracted from the subjects' first level fMRI analysis) were calculated for each participant by averaging the mean signal within the whole left and right VS, HP, and SN/VTA ROIs (created independently without using any result from the word-learning task, see previous section) for each condition of interest (M+ correct first sentence, M+ incorrect first sentence, M- correct first sentence, M- incorrect first sentence, M+ correct second sentence, M+ incorrect second sentence, M- correct second sentence, M- incorrect second sentence). Our hypothesis was that striatal, midbrain, and hippocampal ROIs would show enhanced activation during successful word-learning due to the intrinsic reward produced by learning the meaning of a new-word. Thus, we focused on a triple interaction between Order, Condition, and Response that would reveal that the areas of interest are only activated when participants learn a new-word. For significant interactions related to these factors, two-tailed paired t-tests for all Correct vs. Incorrect conditions and ROIs were planned. An FDR p<0.05 threshold was used to account for multiple testing as 6 ROIs (left and right VS, HP, and SN/VTA) and 4 paired tests per ROI were calculated (M+ first sentence Correct vs. M+ first sentence Incorrect, M- first sentence Correct vs. M- first sentence Incorrect, M+ second sentence Correct vs. M+ second sentence Incorrect, M- second sentence correct vs. M- second sentence Incorrect). This procedure was repeated for the control ROI covering the left and the right primary visual cortex. In this case, individual beta coefficients were submitted to a $2 \times 2 \times 2 \times 2$ repeated measures ANOVA with the factors Hemisphere (Left, Right), Order (first sentence, second sentence), Condition (M+, M-), and Response (Correct, Incorrect).

## ROI analysis II: memory effects

In order to characterize possible long-lasting memory effects within the areas of interest, a second event-related design matrix was specified also using SPM8. M+ correct trials were divided among those in which subjects learned the new-word inside the scanner and still remembered it in the test carried out after the encoding session (remembered condition) and those in which the new-word was not correctly identified in the post-scan test (forgotten condition; see for a similar approach *Gruber et al., 2014*). Trial onsets were modeled again at the moment of the presentation of the new-word. Participants with fewer than three trials within the forgotten or remembered condition were excluded. Hence, M+ correct trials were further subdivided and four new conditions were specified: M+ remembered first sentence, M+ forgotten first sentence, M+ remembered second sentence, M+ forgotten second sentence. After model estimation, first-level contrasts were specified for all participants using each condition against the implicit baseline.

Another full-factorial ROI analysis was calculated, in the same fashion as the one described above to control for novelty and task-difficulty. Individual mean beta coefficients were extracted from all participants by averaging the mean signal within the ROIs of interest (created independently without using any result from the word-learning task, see previous sections) for each of the four aforementioned conditions (M+ remembered first sentence, M+ forgotten first sentence, M+ remembered second sentence, M+ forgotten second sentence). These beta coefficients were submitted into a $2 \times 2 \times 3 \times 2$ repeated measures ANOVA with the factors Hemisphere (Left, Right), Order (first sentence, second sentence), ROI (VS, HP, SN/VTA), and Memory (Remembered, Forgotten). Specifically, we focused on an Order $\times$ Memory interaction that might show greater activation within the areas

of interest for remembered compared to forgotten items, only during second sentence presentation. Two-tailed paired t-tests for all remembered vs. forgotten conditions and ROIs were planned in case the Order × Memory interaction was significant. An FDR p<0.05 threshold was used to account for multiple testing as six ROIs (left and right VS, HP, and SN/VTA) and two paired tests per ROI (M+ first sentence Remembered vs. M+ first sentence Forgotten, M+ second sentence Remembered vs. M+ second sentence Forgotten) were calculated. The exact same procedure was repeated for the incongruent condition, in order to calculate the Order × Memory interaction for M- remembered and forgotten trials. In addition, the same procedure was also repeated for the control ROI covering the left and the right primary visual cortex. In this case, individual beta coefficients were submitted to a 2 × 2 × 2 repeated measures ANOVA with the factors Hemisphere (Left, Right), Order (first sentence, second sentence) and Memory (Remembered, Forgotten).

## Interregional functional connectivity analysis

The physiological connectivity among brain regions varies with the psychological context (*Friston et al., 1997*). We used a psychophysiological interaction (PPI; *Friston et al., 1997*) analysis to assess whether connectivity changes between the VS, the HP, and the VTA/SN in the context of word-learning were predictive of memory success (*Ofen et al., 2012*; *Passamonti et al., 2009*; *Cremers et al., 2010*).

Four mm radius spheres were constructed around the maxima obtained for the left and right VS, HP, and SN/VTA in the ROI analysis. For all participants, individual deconvolved time-series were extracted from all voxels within the left and right spheres. The element by element product of the extracted time-series (the first eigenvariate from every voxel in the sphere) and a vector that coded the main effect of task (1 for M+ correct remembered second sentence, −1 for M+ incorrect second sentence, 0 for the remaining conditions) was then calculated. The result of this product was then reconvolved with the canonical HRF to create the final PPI regressor. For each individual, six extended SPM8 models were built (one for the left and right VS, HP, and SN/VTA). The model included the conditions previously defined for the remembered vs. forgotten analysis, the movement parameters, the deconvolved time-series, and the derived PPI as regressors. Individual models were estimated and main contrasts were generated to test the effects of the PPIs regressor. The computed first level PPI contrast images were entered into a second level random effect analysis (one-sample t-test) which included a covariate with the percentage of correctly remembered M+ new-words during the encoding phase that were still correctly recognized during the post-scan test (remembered new-words). These analyses were restricted to the SN, HP, and SN/VTA ROIs and to the extra control ROI covering the primary visual cortex. Thus, we identified areas within our ROIs that showed connectivity correlating positively or negatively with the subject-specific percentage of remembered words. In order to further prove the specificity of our results to M+ remembered trials, these aforementioned procedures was repeated first by focusing on general effects of condition (all M+ vs. all M- trials) and then on the M- remembered vs. M- incorrect contrast. All statistical maps were thresholded at a p<0.005 uncorrected threshold with ten voxels of cluster extent. Only clusters with peaks surviving a p<0.05 FWE small-volume correction are reported. Maxima and all coordinates are reported in MNI space.

## Word-learning task for exp. 2

### Experimental task

Stimuli were presented using the Psychophysics Toolbox 3.09 (*Brainard, 1997*) and Matlab version R2008b (7.7.0, 32 bit). The structure of the task was virtually identical to that of the main fMRI experiment. Material consisted of 80 pairs of eight word-long Spanish sentences ending in a new-word, built with an increasing degree of contextual constraint (*Mestres-Missé et al., 2009*). The new-words respected the phonotactic rules of Spanish and always stood for a noun (mean frequency 38.62 per million). Forty M+ and forty M- pairs of sentences were presented during 10 consecutive runs following the same procedure as in Exp. 1 (no NR sentences were used). Three differences exist between the fMRI and this behavioural word-learning experiment. First, the recognition test, which followed the same structure of the test performed after Exp. 1 (chance level was 33%: no consistent meaning, consistent meaning on the left, consistent meaning on the right), was completed approximately 24 hr after the encoding phase. Second, since head movements and on-line voice recording

was no longer an issue (the fMRI set-up did not allow for this feature), participants were now instructed to produce a verbal answer 8 s after the new-word of a second sentence appeared. If participants thought that the new-word had a congruent meaning, they had to provide its meaning in Spanish (e.g., *graveyard*). If the new-word had no consistent meaning, they had to say the word *incongruent*. If they did not know whether the new-word had a consistent meaning or not, they had to remain silent. Vocal answers were recorded and later corrected (for the M+ condition, incorrect answers included misses, providing the wrong meaning or saying *incongruent;* for the M- condition, incorrect answers included misses or providing any meaning at all). No recognition tests were performed after each of the encoding runs since there was no longer need for it. And third, participants had to rate their emotions with respect to arousal and pleasantness using the 9-point visual *Self-Assessment Manikin* scale (SAM). For valence/pleasantness, the SAM ranges from a sad, frowning figure (i.e., very negative) to a happy, smiling figure (i.e., very positive). For arousal, the SAM ranges from a relaxed figure (i.e., very calm) to an excited figure (i.e., very aroused). Thus, after being prompted to provide a verbal answer, subjects were requested to enter, using the keyboard, a value between −4 and 4 (9 point scale with 0 as the neutral value). To avoid biasing our results, participants were not told at any point prior to the start of the experiment that the goal of the study was to assess whether the learning of a new-word and its meaning was intrinsically rewarding. Instead, they were told that the objective of the study was to assess how reading load affects mood and that, in order to ensure that there was a real reading load, they had to learn the words of the M+ condition and to detect the incongruence of the new-words from the M-. Finally, participants were instructed that they had to give pleasantness and arousal ratings when the second sentences appeared because that moment signaled that reading load had already occurred (i.e., half of the encoding block had already elapsed). After the experiment, participants were first questioned about the objective of the study. After an answer was provided, we asked them whether successful word-learning had been rewarding in any way.

## Electrodermal activity acquisition and data preprocessing

One of the most widely used measures of bodily states of arousal, including emotional processing and reward, is electrodermal activity (EDA; *Boucsein, 2012*; *Dawson et al., 2000*). This index is modulated by the degree of sympathetic activation: increased sweating provokes a reduction in skin resistance which results in an enhancement of EDA (*Dawson et al., 2000*). Variations in EDA can be decomposed into tonic—reflecting baseline sympathetic activity and known as skin conductance level (SCL)—and phasic changes. The later, also called skin conductance responses (SCR), are related to transient changes in physiological arousal, and have been associated (among other processes) to enhanced memory formation, emotional processing, and motivational behavior (*Damasio, 1994*; *Cahill et al., 1998*; *Mas-Herrero et al., 2014*). Moreover, several studies have shown that explicit monetary rewards modulate SCRs, with larger wins generally generating stronger signals than smaller ones (*Lole et al., 2014*, *2012*; *Dixon et al., 2010*; *Goudriaan et al., 2006*; *Wilkes et al., 2010*). Finally, previous research using functional magnetic resonance imaging (fMRI) has shown that SCRs are modulated by a wide neural network including several structures central to reward processing, such as the ventromedial prefrontal and orbitofrontal cortex, the amygdala, and the striatum (*Nagai et al., 2004*; *Critchley et al., 2000*). All in all, this evidence suggests that SCRs can be a good marker of reward processing (*Neuhaus et al., 2015*). Thus, EDA was used as an objective measure of emotional processing during the word-learning task.

Electrodermal activity was recorded using a Brainvision amplifier (BrainAmp ExG) and a galvanic skin conductance (GSR) module. Two silver/silver-chloride (Ag/AgCl) electrodes were filled with an inert electrolyte cream and placed on the volar surface of the distal phalanx of the index and middle fingers of the non-dominant hand. Participants were required to wash their hands with a non-abrasive soap prior to having the electrodes attached (*Dawson et al., 2000*). Skin conductance was recorded at a constant voltage of 0.5 V and sampled at 200 Hz (0.005 s intervals). Since electrodermal activity is comprised of both phasic responses and a slowly drifting tonic component (*Boucsein, 2012*), raw data for each participant was filtered with a first order Butterworth high-pass filter with a cut off frequency of 0.05 Hz, following current recommendations (*Bach et al., 2013*). One participant was excluded due to problems with data collection.

Our analysis focused on the phasic increases in skin conductance following the presentation of a new-word. Single trial SCRs were assessed by subtracting the mean electrodermal activity during the 1000 ms previous to the appearance of a new-word (note that no stimulus is presented during this time, see *Figure 1C*) from the signal recorded during the 8 s posterior to stimulus onset. Previous studies assessing electrodermal activity have shown that SCR during this time-window is modulated by explicit rewards (*Lole et al., 2012*, *2014*). Note that, since verbal responses can influence SCRs (*Rothen et al., 2014*), participants were instructed to provide an answer after the aforementioned time-window (see *Figure 1C*). First and for each participant, trials were normalized within each condition of interest. Trials in which the SCR signal deviated more than 2 SDs during the first 500 ms after stimulus onset were excluded from the analysis as they might represent artifacts (note that SCRs have a typical latency of 1–3 s; *Dawson et al., 2000*). The average number of rejected trials per condition was $1.93 \pm 0.2$. The number of rejected trials per condition was submitted into a $2 \times 2 \times 2$ repeated measures analysis of variance (ANOVA) with the factors Order (first sentence, second sentence), Condition (M+, M-), and Response (Correct, Incorrect). No significant effects or interactions were found (all $p > 0.27$), indicating that rejected trials were evenly distributed among conditions.

The unnormalized EDA signals of the remaining valid trials associated to our 8 specific conditions (M+ correct first sentence, M+ incorrect first sentence, M- correct first sentence, M- incorrect first sentence, M+ correct second sentence, M+ incorrect second sentence, M- correct second sentence, M- incorrect second sentence) were then averaged for each subject. For each participant, the resulting mean SCR values were normalized across conditions (*Ben-Shakar, 1985*; *Mas-Herrero et al., 2014*; *Packard et al., 2014*). Based on previous studies showing that SCR signals peak around 3–4 s after stimulus onset (*Lole et al., 2012*, *2014*; *Dawson et al., 2000*), normalized SCR values were then averaged from seconds 4 (three seconds after the new-word disappears, see *Figure 1*) to 8. Thus, for each participant, we obtained one mean normalized SCR value for each of the eight conditions of interest. Subsequent statistical analyses were performed in SPSS version 18.0.

## SCR analysis: controlling for novelty, attention and cognitive load

We hypothesized that the intrinsic reward obtained from self-monitoring of correct performance while successfully learning the meaning of a new-word would enhance EDA signals. However, several studies have demonstrated that SCRs can also be modulated by novelty, attention, or cognitive load (*Fowles, 1986*; *Boucsein, 2012*; *Barry, 1996*; *Ben-Shakhar, 1994*; *Leal et al., 2008*; *Dawson et al., 2000*). Nevertheless, as in the fMRI study, the design of our paradigm allows us to control for these effects by including the incongruent (M-, no meaning extraction) condition and the order of presentation (first or second sentence) in our analyses.

In order to control for the aforementioned effects, individual mean normalized SCR values were submitted to a $2 \times 2 \times 2$ repeated measures ANOVA with the factors Order (first sentence, second sentence), Condition (M+, M-), and Response (Correct, Incorrect), just as in the fMRI experiment. We focused on a triple interaction between Order, Condition, and Response that might show that SCR signals were only modulated when participants learned a new-word. For significant interactions, we conducted two-tailed paired t-tests comparing the effect of correct responding (M+ first sentence Correct vs. M+ first sentence Incorrect, M- first sentence Correct vs. M- first sentence Incorrect, M+ second sentence Correct vs. M+ second sentence Incorrect, M- second sentence correct vs. M- second sentence Incorrect). An FDR $p < 0.05$ threshold was used to account for multiple testing.

## SCR analysis: long-term memory effects

In order to characterize whether SCRs were also sensitive to long-term memory effects, M+ correct trials were divided among those in which subjects learned a new-word during the encoding phase and still remembered it in the test carried out 24 hr later (remembered condition) and those in which the new-word was not correctly identified in the follow-up test (forgotten condition), just as in the fMRI analysis. Participants with fewer than three trials within the forgotten or remembered condition were excluded. M+ correct trials were subdivided and four new conditions were specified: M+ remembered first sentence, M+ forgotten first sentence, M+ remembered second sentence, M+ forgotten second sentence. Another full-factorial analysis, in the same fashion as the one described above, was calculated. Individual mean normalized SCR values were submitted to a $2 \times 2$ repeated

measures ANOVA with the factors Order (first sentence, second sentence) and Memory (Remembered, Forgotten). Specifically, we focused on an Order × Memory interaction that might show that SCRs were enhanced for remembered compared to forgotten items only during second sentence presentation. For significant interactions, we conducted two-tailed paired t-tests comparing the effect of correct responding (M+ first sentence Remembered vs. M+ first sentence Forgotten, M+ second sentence Remembered vs. M+ second sentence Forgotten). An FDR p<0.05 threshold was used to account for multiple testing. The exact same procedure was repeated for the incongruent condition, in order to calculate the Order × Memory interaction for M- remembered and forgotten trials.

## Statistical analysis for pleasantness and arousal scales

Ratings for pleasantness and arousal scales (SAM scales; *Bradley and Lang, 1994*) were first submitted into two 2 × 2 repeated measures ANOVAs with the factors Condition (M+, M-) and Response (Correct, Incorrect). We focused on an interaction between Condition and Response that might show that ratings were higher during successful word-learning. For significant interactions, we conducted two-tailed paired t-tests comparing the effect of correct responding (M+ Correct vs. M+ Incorrect and M- Correct vs. M- Incorrect). An FDR p<0.05 threshold was used to account for multiple testing. Regarding the 24 hr recognition test and following the procedure applied to SCRs for long-term memory effects, M+ correct trials were divided among those in which subjects learned the new-word on the previous day and still remembered it in the test carried out 24 hr after the encoding session (remembered condition) and those in which the new-word was not correctly identified in the follow-up day (but still learned on the encoding session; forgotten condition). One paired t-test was used to compare whether ratings for arousal and pleasantness were greater for remembered than for forgotten M+ new-words. The exact same procedure was repeated for the incongruent condition, in order to calculate possible memory effects related to M- remembered and forgotten trials.

## Word-learning task for experiment 3

### Experimental task

Stimuli were presented using the Psychophysics Toolbox 3.0.10 (*Brainard, 1997*) and Matlab version R2008b (7.7.0, 32 bit). The structure of the task was virtually identical to that of Exp. 2 (using the same stimuli, task structure, task instructions, number of trials and timings). The main objectives of Exp. 3 were: (i) explore participants' internal and subjective evaluation of correct performance (especially for false alarms in the M- condition) and (ii) replicate behavioral effects for even longer retention intervals (1 week). Thus, two main differences exist between Experiments 2 and 3. First, before rating their emotions with respect to arousal and pleasantness, participants also provided a confidence rating that allowed us to assess the subjective evaluation of their performance. For consistency reasons, the same scale as for pleasantness and arousal ratings (ranging from −4 for very unsure to 4 for very sure, with 0 as a neutral value) was used. And second, the recognition test was completed both 24 hr and 1 week after encoding and also included a Not Remembered option. Thus, the chance level was at 25% (no consistent meaning, consistent meaning on the left, consistent meaning on the right, not remembered). Besides that, the structure of the recognition test at both 24 hr and 1 week was the same as in Exp. 2. In this case and, taking into account that participants had to come three times to the behavioral laboratory (encoding phase, recognition after 24 hr, recognition after 1 week), we opted to make subjects aware than on sessions 2 and 3 a recognition test on both M+ and M- new-words would be completed.

### Statistical analysis for pleasantness, arousal and confidence scales on exp. 3

As in Exp. 2, ratings for pleasantness, confidence and arousal scales were first submitted into two 2 × 2 repeated measures ANOVAs with the factors Condition (M+, M-) and Response (Correct, Incorrect). We focused on an interaction between Condition and Response that might show that ratings were higher during successful word-learning. For significant interactions, we conducted two-tailed paired t-tests comparing the effect of correct responding (M+ Correct vs. M+ Incorrect and M- Correct vs. M- Incorrect). An FDR p<0.05 threshold was used to account for multiple testing. Taking into account that one of the main objectives of Exp. 3 was to assess participants' internal

evaluation of correct performance with a special emphasis on M- false alarms (i.e., a participant provided a meaning for an M- new-word when he/she should have said *incongruent*) rating scales for incorrect answers were separated into different categories. M- incorrect were divided into false alarms and misses (no answer was provided at all). M+ incorrect trials were divided into incorrect-incongruent (saying incongruent when a meaning should have been provided), incorrect-meaning (a wrong meaning was provided) and misses (no answer). Taking into account that behavioral results for Exp. 2 show that the most common type of incorrect trials were, for M- trials the false alarms (30% of total cases; 73% of incorrect M- trials) and for M+ the incorrect-incongruent trials (24% of total cases; 63% of incorrect M- trials) for ANOVA analyses these were the only type of trials included in the incorrect category.

As in Exp 2., regarding the 24 hr recognition test, M+ correct trials were divided among those in which subjects learned the new-word on the previous day and still remembered it in the test carried out 24 hr after the encoding session (remembered condition) and those in which the new-word was not correctly identified in the follow-up day (but still learned on the encoding session; forgotten condition). The exact same procedure was repeated for the incongruent condition, in order to calculate possible memory effects related to M- remembered and forgotten trials. For the 1 week recognition test, the same procedure was employed with the further constrain that remembered M+ and M- new-words 7 days after encoding had to be remembered in both tests (i.e., a new-word was only considered as remembered after 1 week if it had been previously correctly identified at the 24 hr test). One paired t-test was used to compare whether ratings for confidence, arousal and pleasantness were greater for remembered than for forgotten M+ and M- new-words.

## Acknowledgements

We thank T Pohl, D Scheermann, and K Moehring for their help scanning the participants and Alex Waite for proofreading this manuscript. The present project has been funded by the Spanish Government (MINECO Grants PSI2011-29219 to ARF, PSI2012-37472 to JMP, and FPU program AP2010-4179 to PR) and the Deutsche Forschungsgemeinschaft (SFB779/TPA15 to TN).

## Additional information

### Funding

| Funder | Grant reference number | Author |
| --- | --- | --- |
| Ministerio de Educación, Cultura y Deporte | PSI2011-29219 | Antoni Rodriguez-Fornells |
| Ministerio de Educación, Cultura y Deporte | PSI2012-37472 | Josep Marco-Pallarés |
| Ministerio de Educación, Cultura y Deporte | AP2010-4179 | Pablo Ripollés |
| Deutsche Forschungsgemeinschaft | SFB779/TPA15 | Toemme Noesselt |

The funders had no role in study design, data collection and interpretation, or the decision to submit the work for publication.

### Author contributions

PR, Conception and design, Acquisition of data, Analysis and interpretation of data, Drafting or revising the article; JM-P, AR-F, TN, Conception and design, Analysis and interpretation of data, Drafting or revising the article; HA, CT, Acquisition of data, Drafting or revising the article

### Author ORCIDs

Toemme Noesselt, http://orcid.org/0000-0002-9611-9713

### Ethics

Human subjects: This study was performed according to local ethics (Bellvitge's Hospital Ethical Comitee). All participants gave informed written consent and received compensation for their participation in the study.

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

## Appendix 1

### Word-learning task for fmri experiment 1

#### Behavioral results

The thirty-six participants correctly recognized (mean, std) $60 \pm 15\%$ of new-words from the M+ condition during encoding, as stated in the main text. For this condition, in $31 \pm 14\%$ of the cases participants incorrectly pressed the button in their left hand (mistakenly indicating that the new-word being tested had no congruent meaning). In the remaining $9 \pm 8\%$ of the test trials, they chose the incorrect meaning that served as a foil (it was the meaning evoked by another second sentence not related to the new-word being tested). For the M- condition, as indicated in the main text, participants correctly indicated an absence of coherent meaning in $61 \pm 22\%$ of the cases during the encoding phase. In this case, in $29 \pm 17\%$ of the cases, participants incorrectly selected the meaning evoked by the second sentence of the new-word being tested (e.g., *coffee* when testing *Heutil,* see **Figure 1A**). In the remaining $10 \pm 9\%$ of M- test trials, participants chose the other incorrect meaning that also served as a foil but that was not related to the new-word being tested (it was the meaning evoked by another second sentence of the run).

### Word-learning task for experiment 2

#### Behavioral results

As stated in the main text, participants ascribed, on average, correct meaning to $62 \pm 16\%$ of new-words from the M+ condition during encoding. In $24 \pm 12\%$ of M+ trials participants mistakenly indicated that the new-word being tested had no congruent meaning (using the word *Incongruent*), while in $6 \pm 5\%$ of cases they provided an incorrect meaning. In the remaining $8\% \pm 12\%$ of M+ trials no answer was provided. In $59 \pm 16\%$ of the M- trials participants correctly indicated an absence of coherent meaning. In $30 \pm 16\%$ of the cases, a meaning was incorrectly provided, while in the remaining $11 \pm 15\%$ of M- trials no answer was given. The pattern of results from all experiments (including Exp. 3, see below) was remarkably similar. Indeed, when submitting the number of learned new-words (M+) and the number of correctly rejected new-words (M-) to a mixed repeated measures ANOVA with Condition (M+, M-) as a within-subjects variable and Group (Exp. 1: fMRI participants, Exp. 2:Behavioral-EDA participants, Exp. 3: Behavioral participants) as a between subjects variable, no significant effect of Group [$F(2,81) = 0.09$, p>0.91, partial η2 = 0.002] or Group $\times$ Condition interaction was found [$F(2,81) = 0.654$, p>0.52, partial η2 = 0.016].

### Word-learning task for experiment 3

#### Behavioral results

Participants ascribed, on average, correct meaning to $57 \pm 14\%$ of new-words from the M+ condition during encoding. In $29 \pm 13\%$ of M+ trials participants mistakenly indicated that the new-word being tested had no congruent meaning (using the word *Incongruent*), while in $6 \pm 7\%$ of cases they provided an incorrect meaning. In the remaining $8\% \pm 8\%$ of M+ trials no answer was provided. In $62 \pm 15\%$ of the M- trials participants correctly indicated an absence of coherent meaning. In $25 \pm 13\%$ of the cases a meaning was incorrectly provided, while in the remaining $13 \pm 15\%$ of M- trials no answer was given. When asked about the purpose of the study, most subjects answered that it was to assess their ability to learn new-words (note that in this experiment participants knew that they were going to be tested 24 hr and seven

days after encoding). None of them answered that the specific purpose of the study was to assess the effect of reward on learning (or similar). Nevertheless, all subjects did state that successful learning of a new-word was rewarding.

Importantly, in the recognition test carried out 23.79 ± 2.30 hr after encoding, participants still recognized the correct meaning of 55 ± 18% of learned new-words during the encoding phase, significantly above chance level [$t(23)$ = 8.17, p<0.001, d = 1.61; chance level was set at 25%, see above Materials and methods for Exp. 3]. Note that, as expected, participants only recognized the correct meaning of 19 ± 14% of M+ new-words which were not learned during encoding [significantly below the recognition rate for learned M+ new-words, $t(23)$ = 8.01, p<0.001, d = 2.20]. Regarding the incongruent condition, participants correctly indicated that 35 ± 19% of M- new-words identified during the encoding phase had no meaning ascribed in the 24-hr test, significantly above chance level [$t(23)$ = 2.46, p<0.022, d = 0.48]. However, the 24-hr recognition rate for M- new-words which were not identified during the encoding phase was 28 ± 19%, which is not significantly different from the 24-hr recognition rate for M- new-words correctly identified during encoding [$t(23)$ = 1.8, p>0.085, d = 0.34].

In the test completed approximately 1 week after the encoding session (7 days and 4.97 ± 15 hr; one participant dropped out from the study) subjects still recognized the meaning of 40 ± 15% of learned new-words during the encoding phase (a new-word was considered remembered if it was correctly identified in both the test at 24 hr and the test at one week), significantly above chance level [$t(22)$ = 5.16, p<0.001, d = 1.03]. Regarding the incongruent condition, participants correctly indicated that 16.49 ± 17% of M- new-words identified during the encoding phase had no meaning ascribed in the 1-Week test (a M- new-word was considered remembered if both at the test at 24 hr and at 1 week a participant correctly indicated that the word had no meaning attached), which is below chance level. This shows that at longer intervals participants were still able to remember the new-words learned during the M+ trials.

## Statistical analysis for pleasantness, arousal and confidence scales for exp. 3

We found a significant interaction of Condition × Response for pleasantness [$F(1,23)$ = 14.93, p<0.001, partial η2 = 0.394] but not for arousal ratings [$F(1,23)$ = 1.69, p>0.2,, partial η2 = 0.069, see **Figure 7**]. A significant interaction was also found for the confidence ratings [$F(1,23)$ = 25.74, p<0.001, partial η2 = 0.528]. Two-tailed paired t-test comparisons revealed that pleasantness ratings after correct versus incorrect trials were higher for the M+ [$t(23)$ = 5.01, p<0.001, d = 0.94, FDR-corrected], but not for the M- condition [$t(23$ = −0.41, p>0.73, d = −0.05]. Confidence ratings were also higher for correct versus incorrect trials only for the M+ condition [M+: $t(23)$ = 8.16, p<0.001, d = 1.76; M-: $t(23)$ = 0.09, p>0.92, d = 0.02]. Thus, high pleasantness and confidence ratings were associated only with successful word-learning, which replicates results from Exp. 2. Important to our hypothesis, confidence ratings for M- false alarms (i.e., providing a meaning for an incongruent sentence), were much lower than for M+ correct trials. As a matter of fact, confidence ratings were higher for M+ correct trials than for any other condition (all ps<0.001; see **Figure 7**).

Concordantly, subjective pleasantness and confidence ratings were also higher for remembered than for forgotten M+ new-words in the 24-hr recognition test [$t(22)$ = 4.58, p<0.001, d = 0.61; $t(22)$ = 5.34, p<0.001, d = 0.77, respectively; one subject was excluded from this analysis as had fewer than 3 remembered trials], while no difference in arousal ratings was found [$t(22)$ = 1.65, p>0.11, d = 0.12]. These results suggest again that intrinsic reward derived from internal evaluation of learning success had a modulatory effect on long-term memory and replicates the effects of Exp. 2. Regarding the M- condition, there was no difference in subjective pleasantness [$t(20)$ = 0.66, p>0.51, d = 0.07], arousal [$t(20)$ = 0.70, p>0.48, d = 0.04] or confidence ratings [$t(20)$ = 0.82, p>0.41, d = 0.10] for M- new-words which were correctly identified during the encoding phase and still correctly rejected in the 24-hr test (i.e., subjects selected the incongruent option; M- remembered) and those which

were not correctly identified in the following test (i.e., subjects incorrectly selected one of the two possible meanings or pressed the *Not Remembered* option; M- forgotten). Three subjects were excluded from this analysis as they did not correctly identify in the 24-hr test any of the M- new-words correctly identified during the encoding phase. These results show, as expected, that neither pleasantness nor arousal scales, nor confidence ratings were modulated by memory effects related to the M- condition.

Finally, and most importantly, subjective pleasantness and confidence ratings were also higher for remembered than for forgotten M+ new-words in the 1-week recognition test [$t(22) = 4.14$, $p<0.001$, $d = 0.49$; $t(22) = 5.09$, $p<0.001$, $d = 0.72$, respectively; one subject dropped out from the study and was excluded from the analysis], while no difference in arousal ratings was found [$t(22) = 1.15$, $p>0.26$, $d = 0.10$]. These results suggest again that intrinsic reward derived from internal evaluation of learning success had a modulatory effect on long-term memory even at longer retention intervals. For the M- condition, there was no difference in subjective pleasantness [$t(18) = 1.15$, $p> 0.26$, $d = 0.19$], arousal [$t(18) = 0.69$, $p>0.49$ $d = 0.11$] or confidence ratings [$t(18) = 0.60$, $p>0.55$, $d = 0.11$] for M- new-words which were correctly identified during the encoding phase and still correctly rejected in the 24-hr test and in the 1-week test (i.e., subjects selected the incongruent option; M- remembered) and those which were not correctly identified in the follow-up test (i.e., subjects incorrectly selected one of the two possible meanings or selected the *Not Remembered* option; M- forgotten). These results show, once more, that neither pleasantness nor arousal scales, nor confidence ratings were modulated by memory effects related to the M- condition at a longer retention period.

Ripollés *et al*. eLife 2016;5:e17441. DOI: 10.7554/eLife.17441

## Appendix 2

### Word-learning task for fmri experiment 1

#### ROI fMRI analysis: memory effects for the M+ condition controlling for the number of trials

As the number of remembered trials doubled those of the forgotten condition, a second analysis was performed to rule out the possibility that the memory effects were simply caused by the higher number of remembered events. For this last analysis, only those participants for whom the difference between the numbers of remembered and the number of forgotten trials was fewer than 4 were selected. Data from nine subjects [average number of trials with standard deviation for each condition: $11.33 \pm 3.46$ remembered trials and $9.45 \pm 3.64$ forgotten trials; $t(8) = 2.16$, p>0.0625, d = 0.50, no significant differences in number of trials between conditions] were submitted to the repeated measures ANOVA. For this sub-analysis, the Order $\times$ Memory interaction was still significant [$F(1,8) = 7.708$, p<0.024, partial $\eta2 = 0.491$] and was not affected by region or hemisphere (all ps>0.466). Concordantly, subsequent paired t-test comparisons showed again significant differences in remembered versus forgotten trials in the left and right VS, HP, and SN/VTA during second sentence presentation [left VS, $t(8) = 4.14$, p<0.004, d = 1.44; right VS, $t(8) = 4.65$, p<0.002, d = 0.92; left HP, $t(8) = 3.37$, p<0.01, d = 1.11; right HP, $t(8) = 3.06$, p<0.016, d = 1.19; left SN/VTA, $t(8) = 3.07$, p<0.016, d = 1.19; right SN/VTA, $t(8) = 3.47$, p<0.009, d = 1.36; two-tailed, all reported p-values survived a p<0.05 FDR-correction that accounted for the multiple paired t-tests calculated]. This last analysis strongly suggests that the effect of reward in word-learning for remembered words was not due to the greater number of trials in the remembered condition.

## Appendix 3

### Word-learning task for experiment 2

#### EDA: controlling for novelty, attention, and cognitive load using the whole time-course

Results for EDA analyses were calculated again using the mean normalized SCR signal for the whole time-course (a total of 8 s) instead of using the average between seconds 4 and 8 as in the main text. All reported results in the main text are still significant when using the whole time-course. Therefore, the observed effect appears to be robust and does not depend upon the selection of a particular temporal window. A significant interaction of Order × Condition × Response [$F(1,22) = 12.036$, $p<0.002$, partial $\eta2 = 0.354$] was found. Concordantly, two-tailed paired t-test comparisons for all correct versus incorrect conditions showed significant differences after FDR correction only for M+ correct versus incorrect trials during second sentence presentation [$t(22) = 4.33$, $p<0.001$, $d = 0.84$; $p>0.31$ for all other comparisons].

#### EDA: long-term memory effects for the M+ condition using the whole time-course

Results for EDA analyses were calculated again using the mean normalized SCR signal for the whole time-course (a total of 8 s) instead of using the average between seconds 4 and 8 as in the main text. All reported results in the main text are still significant when using the whole time-course. We found a significant interaction of Order × Memory [$F(1,20) = 8.625$, $p<0.008$, partial $\eta2 = 0.301$]. Two-tailed paired t-test comparisons for all correct versus incorrect conditions showed significant differences after FDR correction for M+ remembered versus forgotten trials during second [$t(20) = 3.66$, $p<0.002$, $d = 0.80$], but not during first [$t(20) = 0.02$, $p>0.98$, $d = 0.006$] sentence presentation.

