## [Decision Letter]

Thank you for submitting your article "Intrinsic monitoring of learning success facilitates memory encoding via the activation of the SN/VTA-Hippocampal loop" for consideration by *eLife*. Your article has been reviewed by three peer reviewers, including Kathryn Dickerson (Reviewer #2), and the evaluation has been overseen by a Reviewing Editor and Sabine Kastner as the Senior Editor.

The reviewers have discussed the reviews with one another and the Reviewing Editor has drafted this decision to help you prepare a revised submission.

Summary:

Ripollés and colleagues report on a study investigating interactions between the dopaminergic midbrain, hippocampus, and ventral striatum, a circuit implicated in reward processing, while learning the definitions of novel words in the absence of reward incentives. The authors find engagement of this circuitry during learning during their task which did not involve either reward incentives or feedback cues, Further, memory for word pairs was associated with both self-report measures of pleasantness and physiological arousal. Thus, the others explore a novel domain in which mesolimbic-hippocampal interactions may support episodic memory.

Essential revisions:

1) Generally, the authors need to rework their Introduction and Discussion to more accurately portray their findings within the larger literature. This needs to be expanded upon in 4 different domains. (i) The authors should better integrate their findings with their previous paper (Ripollés, 2014), as this was a reanalysis of the prior dataset (which should be highlighted in the text). Specifically, how do they believe the mechanisms reported in the current manuscript differ from those described in the prior manuscript, as they all show similar patterns? (ii) More attention is warranted describing these results in relationship those reported by Gruber et al. (2014), Duncan, Tompary, & Davachi (2014), and Murty, Dubrow, & Davachi (2015), all of which show engagement of this network during memory encoding in the absence of reward feedback. More attention to the convergence and divergence of these studies and the current study is warranted. (iii) There was concern in framing the paper as 'intrinsic reward'. These claims are made by the absence of feedback as well as engagement of the mesolimbic-hippocampal circuitry. However, there are multiple interpretations of how this circuit could be engaged in the absence of feedback outside of intrinsic reward (i.e., goal-achievement, general positive affect, general memory encoding). I believe it is fine that this may be one factor driving the current results, but these claims need to be qualified. (iv) It is an overstatement to say that 'little is known about the neurobiological mechanisms subserving memory formation when learning is not driven by explicit/external rewards or feedback" as there is a whole field investigating these processes during incidental memory encoding.

2) There were concerns on the selection of ROIS and implementation of ROI-based analyses. The reviewers were enthusiastic about the use of 'neuro-synth' to have an unbiased selection procedure, but why did this deviate from the author's prior work using a functional localizer. Similarly, it was inconsistent when the authors used voxel-wise analysis Similar selection of the regions of interest. I believe their approach of using neuro-synth is an unbiased method to explore the SN/VTA-hippocampus circuit. However, the prior manuscript used a functional ROI. Can the authors justify this decision, versus their prior methods for ROI selection? Similarly, the authors use a voxel-wise analyses within some analyses, and average activity across the entire ROI for others. The reviewers believe this latter approach is more sound, but more importantly need justification for why different methods were using. Further, additional detail on the voxel-wise approach would help clarify these choices.

3) These results provide a sound characterization of how hippocampal-SN/VTA interactions supporting memory encoding in this contexts. However, additional analyses are necessary to help detail the specificity of these findings. Firstly, if the authors choose to only report on ROI based analyses (see below), they should include a control region for both univariate and connectivity analyses, particularly one which may not receive substantial innervation from the SN/VTA (i.e., visual cortex). Alternatively, and preferably, they would include exploratory whole-brain analyses. This is relevant to not only determine the specificity of the findings, but also because theoretical models (including Lisman and Grace) implicate involvement in regions beyond the limited circuit investigated in the current study. This would again be appropriate for both univariate and connectivity analyses. Similarly, additional PPI analyses are warranted detailing the main effect of condition (M+>M-) in addition to difference in memory analyses.

4) Some of the design and analysis considerations need further explanation. (i) Why did the authors choose to do some of their memory tests inside the scanner and others outside the scanner? Were the differences in memory across these two contexts, and if so this needs to be addressed in the discussion. Would it be appropriate to collapse the memory data within the figure? (ii) Was it warranted performing all analyses separately across left and right ROIS as opposed to collapsed bilaterally? If ANOVA did not show effects of laterally, reporting bilateral activations would help streamline the paper. (iii) Justification for the experimental paradigm, as related to the current question of interest, should be emphasized. Why was this paradigm selected versus other paradigms that do not use explicit feedback?

---

## [Author Response]

*Summary:*

*Ripollés and colleagues report on a study investigating interactions between the dopaminergic midbrain, hippocampus, and ventral striatum, a circuit implicated in reward processing, while learning the definitions of novel words in the absence of reward incentives. The authors find engagement of this circuitry during learning during their task which did not involve either reward incentives or feedback cues, Further, memory for word pairs was associated with both self-report measures of pleasantness and physiological arousal. Thus, the others explore a novel domain in which mesolimbic-hippocampal interactions may support episodic memory.*

*We thank Dr. Dickerson, the two other, anonymous, reviewers, the Reviewing Editor and her team and Dr. Kastner as the Senior Editor for their many helpful and very thoughtful comments which we think have helped us greatly to improve the quality of our manuscript. Below we provide detailed answers to each of the reviewers' comments and suggestions.*

*Essential revisions:*

*1) Generally, the authors need to rework their Introduction and Discussion to more accurately portray their findings within the larger literature. This needs to be expanded upon in 4 different domains. (i) The authors should better integrate their findings with their previous paper (Ripollés, 2014), as this was a reanalysis of the prior dataset (which should be highlighted in the text). Specifically, how do they believe the mechanisms reported in the current manuscript differ from those described in the prior manuscript, as they all show similar patterns?*

We thank the reviewers for these crucial comments. Regarding (i), in our previous work we assessed whether successful meaning extraction, in the absence of explicit feedback, could enhance activity within the ventral striatum (VS), a key reward-related region. Word meaning acquisition was assessed directly after initial encoding. However, we never explored if this encoding-related activity, which we hypothesize to be reward-related, had an effect on longer-lasting memory traces (i.e., did this intrinsic reward signal enhance longer lasting memory for the new words that were being encoded?). We have now expanded the section dealing with our previous results to better integrate previous and past research and we have further emphasized that the first part of this work is a re-analysis of previous data.

Introduction, third paragraph:

“Importantly, we recently showed that, in our paradigm, successful meaning extraction enhanced fMRI-signals within the VS. […] We hypothesized that increased brain activity and functional connectivity within the areas of the loop, in absence of any external reward, should be associated with enhanced memory formation (i.e., greater activity and connectivity during encoding for later remembered vs. later forgotten items).”

Discussion, first paragraph:

“This study extends our previous results (Ripollés et al., 2014) by showing that successful learning, in absence of external feedback or reward, engages a complex subcortical network—that includes not only reward, but also memory and dopamine related regions—which seems to modulate the entrance of new information into long-term memory.”

*(ii) More attention is warranted describing these results in relationship those reported by Gruber et al. (2014), Duncan, Tompary, & Davachi (2014), and Murty, Dubrow, & Davachi (2015), all of which show engagement of this network during memory encoding in the absence of reward feedback. More attention to the convergence and divergence of these studies and the current study is warranted. (iii) There was concern in framing the paper as 'intrinsic reward'. These claims are made by the absence of feedback as well as engagement of the mesolimbic-hippocampal circuitry. However, there are multiple interpretations of how this circuit could be engaged in the absence of feedback outside of intrinsic reward (i.e., goal-achievement, general positive affect, general memory encoding). I believe it is fine that this may be one factor driving the current results, but these claims need to be qualified.*

In regard to points (ii) and (iii), we have now expanded the description of the study by Gruber et al., (2014) and have also addressed the alternative interpretations on how the SN/VTA-HP loop might have been engaged in absence of feedback, including the suggestions made by Murty et al., (2014) and Duncan et al., (2015):

Discussion, third paragraph:

“Importantly, not only extrinsic signals but also intrinsic motivational states can enhance memory formation. For example, in a recent study, both the VS and HP showed enhanced activity during the anticipation of trivia answers that were later remembered, only when participants were engaged in states of high curiosity (Gruber et al., 2014). Thus, both anticipation of explicit rewards and intrinsic motivational states can promote memory formation, and both engage the SN/VTA, HP and VS.”

Discussion, fifth paragraph:

“Although our results (coming from fMRI data, EDA signals and subjective pleasantness ratings) suggest that the SN/VTA-HP loop was engaged by an intrinsic reward-related signal that was associated with the participants' monitoring of their learning success, recent research proposes several alternative interpretations on how this circuit could be engaged in the absence of feedback. […] However the output to be learned and the mechanism subserving this learning could still be different: while results in the two aforementioned studies are based on the well-known role of dopamine in the codification of a reward prediction error (Schultz, 1998), we suggest that our learning is driven by the effects that dopamine has in LTP when it is released at the HP (Shohamy and Adcock, 2010).”

(iv) It is an overstatement to say that 'little is known about the neurobiological mechanisms subserving memory formation when learning is not driven by explicit/external rewards or feedback" as there is a whole field investigating these processes during incidental memory encoding.

We have removed the sentence 'little is known about the neurobiological mechanisms subserving memory formation when learning is not driven by explicit/external rewards or feedback" as we agree with the reviewers that it was indeed an overstatement.

Abstract:

Humans constantly learn in the absence of explicit rewards. However, the neurobiological mechanisms supporting this type of internally-guided learning (without explicit feedback) are still unclear.

*2) There were concerns on the selection of ROIS and implementation of ROI-based analyses. The reviewers were enthusiastic about the use of 'neuro-synth' to have an unbiased selection procedure, but why did this deviate from the author's prior work using a functional localizer. Similarly, it was inconsistent when the authors used voxel-wise analysis Similar selection of the regions of interest. I believe their approach of using neuro-synth is an unbiased method to explore the SN/VTA-hippocampus circuit. However, the prior manuscript used a functional ROI. Can the authors justify this decision, versus their prior methods for ROI selection? Similarly, the authors use a voxel-wise analyses within some analyses, and average activity across the entire ROI for others. The reviewers believe this latter approach is more sound, but more importantly need justification for why different methods were using. Further, additional detail on the voxel-wise approach would help clarify these choices.*

We thank the reviewers for their comments and we agree that ROI selection should have been more thoroughly explained in the initial submission. Regarding the reviewers' concerns, as a matter of fact, in both our past (Ripollés et al., 2014) and present work, we used a functional ROI to localize the VS. For the current analyses, we decided to also use a NeuroSynth meta-analysis in order to better pinpoint reward-related regions within our ROIs (to test our hypothesis that word learning is intrinsically rewarding). For this, we first created a set of ROIs by using the functional localizer created in our previous work for the VS (Ripollés et al., 2014) and an anatomical atlas for the HP. For the SN/VTA we specifically acquired proton density images from all our subjects in order to properly identify this structure. Then we used the results from the NeuroSynth meta-analysis on reward to further refine our ROIs. In particular, we masked our original ROIs with the aforementioned meta-analysis. We did this following the work of Gruber and colleagues (2014), which used the same approach (i.e., they created anatomical ROIs of the three regions of the loop and refined them with the NeuroSynth results). Importantly, when we masked the VS ROI from our previous work with the NeuroSynth analysis, the whole original ROI was included. This means that all the voxels included by our functional localizer were also included in the reward meta-analysis. Thus, the selection of VS ROIs in the past and present work are in agreement. Although we indicated this in the previous version of the manuscript [Finally, the three previously created ROIs (VS, HP, and SN/VTA) were masked with this reward mask (as in Gruber et al., 2014; see blue areas in Figure 3 and Figure 4). The VS ROIs previously defined were not affected by the masking, as they overlapped as a whole with the reward mask], we have now included an extended description of ROI selection. We want to apologize for any misunderstandings due to our impoverished previous methods section regarding ROI creation:

Materials and methods, subsection: “ROI creation”:

“We then refined the three previously created ROIs (VS, HP, and SN/VTA) by masking them with the results of the NeuroSynth meta-analysis. […] The original HP ROIs that were created from an anatomical atlas were restricted to the head and anterior body of the hippocampi after masking with the NeuroSynth meta-analysis (number of voxels in the final ROIs: 450 for left HP, 434 for right HP).”

Finally, we agree with the reviewers that including both voxel-wise and average-activity ROI analyses for the first analysis regarding M+ correct vs. incorrect trials is repetitive (former Figure 2 and Figure 3 show the same effects, in a way). We have now removed the voxel-wise GLM analysis (former Figure 2) from the manuscript. In the case of PPI connectivity analyses, we think that using voxel-wise statistics makes a stronger case for our results. In particular, by using voxel-wise analyses we can show whether the areas showing enhanced functional connectivity overlap (e.g., does the same area within the HP show enhanced connectivity when using the VS and the SN/VTA as a source?). In this vein, there was an overlap at the left VS between the analysis in which the seed was placed at the left HP and that in which the left SN/VTA was used as source; the same happened at the HP when using the left SN/VTA or the left VS as seeds (see former Figure 5 or current Figure 4). In addition, using voxel-wise statistic for connectivity analyses is in line with other studies assessing the SN/VTA-HP loop in which average-activity analyses were used for remembered/forgotten comparisons, but voxel-wise statistics were employed for connectivity analyses (e.g. Adcock et al., 2006).

*3) These results provide a sound characterization of how hippocampal-SN/VTA interactions supporting memory encoding in this contexts. However, additional analyses are necessary to help detail the specificity of these findings. Firstly, if the authors choose to only report on ROI based analyses (see below), they should include a control region for both univariate and connectivity analyses, particularly one which may not receive substantial innervation from the SN/VTA (i.e., visual cortex). Alternatively, and preferably, they would include exploratory whole-brain analyses. This is relevant to not only determine the specificity of the findings, but also because theoretical models (including Lisman and Grace) implicate involvement in regions beyond the limited circuit investigated in the current study. This would again be appropriate for both univariate and connectivity analyses. Similarly, additional PPI analyses are warranted detailing the main effect of condition (M+>M-) in addition to difference in memory analyses.*

We thank the reviewers for their suggestion, which we think has provided further support for the specificity of our results. We now include in the manuscript the results for a ROI located at the primary visual cortex (BA17). As expected, activity within this area was not modulated by successful meaning extraction or by memory effects. No connectivity results were found, either. Both the methods and results have been updated to include the new analyses:

Materials and methods section, subsection: “ROI creation”:

“Finally, an extra ROI covering the primary visual cortex (BA17) was created using the toolbox Wfu-pickatlas and the Automated Anatomical Labelling Atlas, as with the HP ROI (Maldjian et al., 2003;Maldjian et al., 2004; Tzourio-Mazoyer et al., 2002). This ROI served as control region to further assess the specificity of our findings (number of voxels in the final ROIs: 362 for left BA17, 370 for right BA17).”

Materials and methods section, subsection: “ROI analysis II: memory effects”:

“This procedure was repeated for the control ROI covering the left and right primary visual cortex. In this case, individual β coefficients were submitted to a 2×2×2×2 repeated measures ANOVA with the factors Hemisphere (Left, Right), Order (1st sentence, 2nd sentence), Condition (M+, M-), and Response (Correct, Incorrect).”

“In addition, the same procedure was also repeated for the control ROI covering the left and right primary visual cortex. In this case, individual β coefficients were submitted to a 2×2×2 repeated measures ANOVA with the factors Hemisphere (Left, Right), Order (1st sentence, 2nd sentence) and Memory (Remembered, Forgotten).”

Materials and methods section, subsection: “Interregional functional connectivity analysis”:

“These analyses were restricted to the SN, HP, and SN/VTA ROIs and to the extra control ROI covering the primary visual cortex.”

Results, third paragraph:

“In addition, no significant Order × Condition × Response [*F*(1,35)=0.354, p>0.55, partial η2=0.010] interaction was found when using a control ROI based at the primary visual cortex (see Materials and methods), which further supports the specificity of our results.”

Same section:

“In addition and, as expected, no significant interaction of Order × Memory was found for the control ROI located at the primary visual cortex for M+ or M- trials t [*F*(1,32)=0.53, p>0.46, partial η2=0.017 and *F*(1,27)=0.023, p>0.88, partial η2=0.001, respectively] which further supports the specificity of our results to reward and memory related regions.”

Results section, subsection: “Is the SN/VTA-HP loop instrumental in enhancing memory formation?”:

“In addition, no significant correlations were found at the control ROI located at the primary visual cortex for any of the seeds.”

Regarding exploratory whole-brain analyses, we already reported in our previous work (Ripollés et al., 2014) the cortical network that was also modulated by successful meaning extraction in our task. This network contained classical language related regions, including the left inferior frontal gyrus, the left inferior parietal gyrus and the left superior medial frontal gyrus. We respectfully believe that including whole brain analyses that cover these regions is beyond the scope of this work, which is clearly focused at the SN/VTA-HP loop. In addition, the cortical language regions that are modulated by successful encoding are also activated (but to a lesser extent) during, for example, first presentation of an M+ or an M- condition. Thus, the detected activity in these language areas is not specific: activity is induced in cortical language-related regions for both M+ and M- conditions, regardless of correctness or order of presentation (it is just that for M+ correct trials, the activity is higher; see for example the first study using a similar type of learning design, Mestres-Missé et al., 2008). However, the SN/VTA-HP loop is only engaged for M+ correct trials at second presentation, when the participants are able to correctly learn the meaning of a new word (i.e., the activity is very specific: no fMRI signals are enhanced within the areas of the loop by first presentation of a new-word, by incorrectly completing the M+ condition or by completing the M- condition regardless of the outcome).

In regards to the additional PPI analyses, we are not completely sure of which specific analyses the reviewers are asking for. In case the reviewers were concerned about whether the reported effect can be attributed to the M+ correct and remembered condition, with no remaining effect attributable to condition-specific main effects, we have computed further analyses that we think might help to improve our Results section. We, however, apologize if this were not the specific analyses that the reviewers requested. First of all in the 2×2×3×2×2 repeated measures ANOVA with the factors Hemisphere (Left, Right), Order (1st sentence, 2nd sentence), ROI (VS, HP, SN/VTA), Condition (M+, M-), and Response (Correct, Incorrect), there was no main effect of condition (p>0.19), further emphasizing that the reported effects were driven by the M+ correct trials. We have now stressed this in the Results section.

Results section, third paragraph:

“Importantly, there was no main effect of condition (p>0.19), further emphasizing that the reported effects were driven by the M+ correct trials.”

Going back to the connectivity results, we computed several additional PPI analyses in order to better characterize the nature of the reported effects. First we repeated the main PPI analyses but this time we focused on enhanced inter-regional coupling driven by main effects of condition (all M+ versus all M- trials). As expected, no significant correlations with the number of remembered words were found, further supporting the specificity of our results to the M+ remembered trials. In addition, no effects were found if the PPI analyses were focused on M- remembered vs. M- incorrect trials. This further provides evidence for the selectivity of the connectivity results to the M+ remembered trials. We have extended the PPI Results section to include these additional results:

Results section, subsection: “Is the SN/VTA-HP loop instrumental in enhancing memory formation?”:

“Finally, we computed several additional PPI analyses in order to better characterize the nature of the reported effects. First we repeated the main PPI calculations but focusing on enhanced inter-regional coupling driven by main effects of condition (all M+ versus all M- trials during second sentence presentation). As expected, no significant correlations with the number of remembered words were found. In the same manner, no effects were found if the PPI analyses focused on M- remembered vs. M- incorrect trials, which further supports the specificity of the connectivity results to the M+ remembered condition.”

*4) Some of the design and analysis considerations need further explanation. (i) Why did the authors choose to do some of their memory tests inside the scanner and others outside the scanner? Were the differences in memory across these two contexts, and if so this needs to be addressed in the discussion. Would it be appropriate to collapse the memory data within the figure?*

We thank the reviewers for raising these important issues and apologize for any misunderstandings. The scanning set-up did not allow for online recording of spoken responses. That is why we decided to ask participants about their performance immediately after each encoding run. Participants completed 10 short encoding runs each lasting less than 3 minutes. Each encoding run was immediately followed by a test. On average, mean time between the second presentation of a new word during the encoding block and the presentation of that very same word during testing was approximately 45 seconds. The objective of these tests was to assess for which new-words from the M+ condition correct meaning had been extracted and which new words from the M- had been correctly rejected during each encoding block. That is why we use the terms *correct* and *incorrect* throughout the manuscript, as this test allows us to assess the performance of our participants during encoding. Although during fMRI Experiment 1 there is an obvious lag (as stated above, 45 seconds on average) between encoding and testing, results from Experiments 2 and 3 were very similar, and these latter Experiments were carried out in an environment in which online responses could be recorded (i.e., participants did not complete a test after each block; instead provided verbal answers 8 seconds after each second presentation of a new word). In particular, for the M+ condition participants learned the meaning of 60%, 62% and 57% of M+ new words in Experiments.1 (fMRI), 2 (EDA) and 3 (behavioral) respectively. The same happens for the control condition in which participants correctly rejected 61%, 59% and 62% of M- new words in Experiments 1, 2 and 3, respectively. The pattern of results from all experiments is remarkably similar. Indeed, as stated in Appendix 1, when submitting the number of learned new-words (M+) and the number of correctly rejected new-words (M-) to a mixed repeated measures ANOVA with Condition (M+,M-) as a within-subjects variable and Group (Experiment1: fMRI participants, Experiment 2: Behavioral-EDA participants, Experiment 3: Behavioral participants) as a between subjects variable, no significant effect of Group [*F*(2,81)=0.09, p>0.91] or Group × Condition interaction was found [*F*(2,81)=0.654, p>0.52]. All in all, these results suggest that the memory test that our participants completed immediately after each encoding block in the fMRI setup was indeed assessing participants’ performance (i.e., for which new words participants had correctly extracted a meaning).

In contrast, in the post-scan test, mean time between encoding (inside the scanner) and testing (after the fMRI session) was 54 minutes. Therefore, this test was assessing which of the learned words were still remembered after almost 1 hour and which had been forgotten. We have now expanded the Materials and methods section to better address this issue:

Materials and methods section, subsection: “Word-Learning Task for fMRI” Exp. 1”:

“As the scanning set-up did not allow for online recording of responses, immediately after each encoding run, participants had to complete a test that was devised to assess their performance (i.e., to assess the words that they had correctly encoded or that they had correctly rejected). On average, mean time between the second presentation of a new word during the encoding block and the presentation of that very same word during testing was approximately 45 seconds.”

“In this post-scan test, mean time between encoding (inside the scanner) and testing (after the fMRI session) of a particular new-word was 54 minutes. Therefore, this test was devised in order to assess which of the learned words inside the scanner were still remembered and which of them had been forgotten after almost 1 hour of retention period.”

(ii) Was it warranted performing all analyses separately across left and right ROIS as opposed to collapsed bilaterally. If ANOVA did not show effects of laterally, reporting bilateral activations would help streamline the paper.

The reviewers raise an important question in point (ii). Although we agree that reporting bilateral activations might ease the flow of the manuscript, we would like to keep the results as they are. The rationale behind this lies in the well-known fact that language is cortically left-lateralized for more than 90% of right handed individuals (Springer et al., 1999). Therefore, researchers from the language domain might find it interesting that subcortical reward and memory related areas show a bilateral pattern, whereas cortical language-related regions were indeed left lateralized during successful word learning in our task as previously reported (Ripollés et al., 2014). We think that showing separated plots in the Figures and separated t-tests in the results section for each ROI further emphasizes this remarkable result (i.e., the fact that reward-related effects were not left-lateralized, even though the items to be memorized were language-related). Of course, if the reviewers still think that the manuscript will benefit from reporting collapsed results for the hemispheres we are willing to further address this question in subsequent reviews.

(iii) Justification for the experimental paradigm, as related to the current question of interest, should be emphasized. Why was this paradigm selected versus other paradigms that do not use explicit feedback?

We selected this task not only because it allows for learning without the need to provide external feedback or guidance, but also (and especially) because it mimics a type of learning that occurs in "real-life" environments and that has been related to vocabulary growth during human lifespan (Nagy, 1985). We have extended the Introduction in order to add this information:

Introduction:

“Related to this, we recently developed a learning task (Ripollés et al., 2014; see also Mestres-Missé et al., 2007) that mimicked our capacity to learn the meaning of new-words presented in verbal contexts, a process that usually occurs without external guidance and that is considered one of the most important sources of vocabulary learning during childhood years (Nagy et al., 1985). […] Therefore, our word-learning task is ideally suited to test internally-guided learning as: i) in our task participants are able to learn the meaning of artificially created new-words by using contextual information, without the need for explicit feedback or reward; and ii) our paradigm mimics an important learning process that occurs in real-world environments.”